# Observation Interference in Partially Observable Assistance Games

**Scott Emmons** [* 1]   **Caspar Oesterheld** [* 2]   **Vincent Conitzer** [2]   **Stuart Russell** [1]

## Abstract

We study partially observable assistance games (POAGs), a model of the human-AI value alignment problem which allows the human and the AI assistant to have partial observations. Motivated by concerns of AI deception, we study a qualitatively new phenomenon made possible by partial observability: would an AI assistant ever have an incentive to interfere with the human's observations? First, we prove that sometimes an optimal assistant must take observation-interfering *actions*, even when the human is playing optimally, and even when there are otherwise-equivalent actions available that do not interfere with observations. Though this result seems to contradict the classic theorem from single-agent decision making that the value of information is nonnegative, we resolve this seeming contradiction by developing a notion of interference defined on entire *policies*. This can be viewed as an extension of the classic result that the value of information is nonnegative into the cooperative multiagent setting. Second, we prove that if the human is simply making decisions based on their immediate outcomes, the assistant might need to interfere with observations as a way to query the human's preferences. We show that this incentive for interference goes away if the human is playing optimally, or if we introduce a communication channel for the human to communicate their preferences to the assistant. Third, we show that if the human acts according to the Boltzmann model of irrationality, this can create an incentive for the assistant to interfere with observations. Finally, we use an experimental model to analyze tradeoffs faced by the AI assistant in practice when considering whether or not to take observation-interfering actions.

## 1. Introduction

Assistance games provide a formalization of the human-AI value alignment problem (Shah et al., 2020). They are based on Hidden Goal MDPs (Fern et al., 2014) and Cooperative Inverse Reinforcement Learning (CIRL) (Hadfield-Menell et al., 2016), an extension of Inverse Reinforcement Learning (IRL) (Ng & Russell, 2000; Abbeel & Ng, 2004). In assistance games, a single human and a single AI assistant share the same reward function, but this reward function is only known to the human; the assistant must learn it. In assistance games, desirable properties, such as teaching by the human and learning by the assistant, *emerge* as optimal solutions to the game (Shah et al., 2020). (This contrasts with prior work on algorithms where teaching is an explicit *objective* (Cakmak & Lopes, 2012; Goldman & Kearns, 1995; Balbach & Zeugmann, 2009).) For example, Woodward et al. (2020) find that deep neural networks solving an assistance game invent strategies that involve information sharing, information seeking, and question answering.

Past analysis of assistance games was done assuming that the state of the world is fully observed by both the human and the assistant (Hadfield-Menell et al., 2016; 2017). While Shah et al.'s (2020) definition of an assistance game allows for partial observability, they do not study its implications. In this work, we introduce the notion of a partially observable assistance game (POAG) to study the more general case faced in reality: when the world is only partially observable. Partial observability raises new issues surrounding the communication of private information. *A priori*, we might hope that AI assistants never take any action that obstructs information. Yet our analysis will show that even assistants which perfectly share our goals must sometimes obstruct information to communicate other, more important information.

This tension connects to broader work on AI deception, which recent research approaches from multiple angles. Park et al. (2024) provide a philosophical definition and empirical survey of AI deception, while Ward et al. (2023) define deception in structural causal games. Of particular relevance is work analyzing how reinforcement learning from human feedback (RLHF)—which can be seen as an algorithm for solving assistance games—can lead to deception. Lang et al. (2024) prove that partial observability in

---
[*]Equal contribution  [1]Center for Human-Compatible AI, University of California, Berkeley [2]Foundations of Cooperative AI Lab, Carnegie Mellon University. Correspondence to: Scott Emmons <emmons@berkeley.edu>.

*Proceedings of the 42nd International Conference on Machine Learning*, Vancouver, Canada. PMLR 267, 2025. Copyright 2025 by the author(s).

RLHF can create dual risks of deceptive inflating and over-justification. Complementing Lang et al. (2024)'s theory, Wen et al. (2024) and Williams et al. (2024) provide experimental evidence that optimizing for human feedback teaches language models to mislead humans. However, these works primarily focus on misaligned AI systems that deceive for their own goals. We study the subtle case where a perfectly aligned AI assistant might obstruct information for the human's benefit.

Concretely, we seek to understand whether observation interference emerges as optimal behavior in an AI assistant that shares the human's goals. We take a game-theoretic approach, studying qualitative properties of *optimal policy pairs* and *best responses* in POAGs. To start, we define an observation interfering action as one which provides the human with a subset of the information available with an otherwise-equivalent action. We then analyze if the AI assistant ever takes observation interfering actions in optimal policy pairs or best responses.

Our analysis reveals three distinct incentives for an AI assistant to take observation interfering actions. First, when the assistant has private information, it might need to interfere with observations to communicate its private information to the human (Section 4.2). This can happen even when the human is playing optimally, and even when there are otherwise-equivalent actions available that do not interfere with observations. This result presents a puzzle, as it seems to contradict the classic theorem from single-agent decision making that the value of information (sometimes also called the value of perfect information) is nonnegative (e.g., Koller & Friedman, 2009, Sect. 23.7; Russell & Norvig, 2010, Sect. 16.6.3). To resolve this seeming contradiction, we develop a notion of interference defined on entire *policies* rather than individual actions. While optimal solutions (i.e., human-AI policy pairs) might involve the AI assistant taking individual actions which would on their own be observation interference, we prove that there is always an optimal solution with no observation interference when we consider the AI assistant's overall policy. This can be viewed as an extension of the classic result that the value of (perfect) information is nonnegative into the cooperative multiagent setting.

This result connects to a broader literature on the value of information in multiagent settings. In games with competing interests, it is well-known that introducing common knowledge can lead to worse outcomes for all players (Kamien et al., 1990). Using a set-theoretic framework, Bassan et al. (2003) establish a class of general-sum games where additional information Pareto-improves all of the Nash equilibria. Their class of games includes common-payoff games. Using a probabilistic framework, Lehrer et al. (2010) extend this analysis to alternative solution concepts. Notably, Bassan et al. (2003) and Lehrer et al. (2010) consider only single-timestep games where players simultaneously act without observing the other players' actions. In our setting, the environment evolves over time, and the players can influence each other's observations through their actions. Our results show that this influence on observations—including observation interference—is a key feature that enables the communication of private information to achieve better outcomes.

In our setting, even if a non-interference solution exists, it might require that the human send information to the assistant via an unnatural communication convention. We find that a second incentive for observation interference occurs if the human is instead just making decisions based on the immediate reward of those decisions. In that case, the assistant's best response might require observation interference as a form of preference query (Section 5). We prove that this incentive for interference goes away if the human is playing optimally, or if we introduce a communication channel for the human to communicate her preferences to the assistant.

When the human is making irrational decisions, it creates a third incentive for the assistant to interfere with observations. For example, we show that if a Boltzmann-rational decision maker has a higher error rate when presented with complete information, the assistant might suppress information to give the human an easier decision (Section 6).

Finally, in Section 7, we use an experimental model to investigate tradeoffs the assistant faces when deciding whether or not to interfere with observations. In line with our theory, we find that observation interference allows the AI assistant to communicate private information, but it comes at the cost of destroying useful information. Measuring this tradeoff, we find that having more private information leads to a stronger incentive to interfere with observations.

## 2. Preliminaries / Setup

### 2.1. Partially Observable Assistance Games

We study partially observable assistance games (Shah et al., 2020):

**Definition 2.1.** *A partially observable assistance game (POAG) $M$ is a two-player DecPOMDP with a human or principal, $\mathbf{H}$, and an AI assistant, $\mathbf{A}$. The game is described by a tuple, $M = \langle \mathcal{S}, \{\mathcal{A}^{\mathbf{H}}, \mathcal{A}^{\mathbf{A}}\}, T(\cdot \mid \cdot, \cdot, \cdot), \{\Theta, R(\cdot, \cdot, \cdot; \cdot)\}, \{\Omega^{\mathbf{H}}, \Omega^{\mathbf{A}}\}, O(\cdot, \cdot \mid \cdot, \cdot, \cdot), P_0(\cdot, \cdot), \gamma \rangle$, with the following definitions: $\mathcal{S}$, a set of world states: $s \in \mathcal{S}$; $\mathcal{A}^{\mathbf{H}}$, a set of actions for $\mathbf{H}$: $a^{\mathbf{H}} \in \mathcal{A}^{\mathbf{H}}$; $\mathcal{A}^{\mathbf{A}}$, a set of actions for $\mathbf{A}$: $a^{\mathbf{A}} \in \mathcal{A}^{\mathbf{A}}$; $T(\cdot \mid \cdot, \cdot, \cdot)$, a conditional distribution on the next world state, given previous state and action for both players: $T(s' \mid s, a^{\mathbf{H}}, a^{\mathbf{A}})$; $\Theta$, a set of possible static reward parameter values, only observed by $\mathbf{H}$: $\theta \in \Theta$; $R(\cdot, \cdot, \cdot; \cdot)$, a parameterized reward function that maps world states, joint actions, and reward parameters to*

*real numbers:* $R : \mathcal{S} \times \mathcal{A}^{\mathbf{H}} \times \mathcal{A}^{\mathbf{A}} \times \Theta \to \mathbb{R}$; $\Omega^{\mathbf{H}}$, *a set of observations for* $\mathbf{H}$: $o^{\mathbf{H}} \in \Omega^{\mathbf{H}}$; $\Omega^{\mathbf{A}}$, *a set of observations for* $\mathbf{A}$: $o^{\mathbf{A}} \in \Omega^{\mathbf{A}}$; $O(\cdot, \cdot \mid \cdot, \cdot, \cdot)$, *a conditional distribution on the observations, given the next world state and action of both players:* $O(o^{\mathbf{H}}, o^{\mathbf{A}} \mid s', a^{\mathbf{H}}, a^{\mathbf{A}})$; $P_0(\cdot, \cdot)$, *a distribution over the initial state, represented as tuples:* $P_0(s_0, \theta)$; *and* $\gamma$, *a discount factor:* $\gamma \in [0, 1]$.

We denote $\mathbf{H}$'s and $\mathbf{A}$'s marginal observation distributions as $O^{\mathbf{H}}(o^{\mathbf{H}} \mid s', a^{\mathbf{H}}, a^{\mathbf{A}}) = \sum_{o^{\mathbf{A}}} O(o^{\mathbf{H}}, o^{\mathbf{A}} \mid s', a^{\mathbf{H}}, a^{\mathbf{A}})$ and $O^{\mathbf{A}}(o^{\mathbf{A}} \mid s', a^{\mathbf{H}}, a^{\mathbf{A}}) = \sum_{o^{\mathbf{H}}} O(o^{\mathbf{H}}, o^{\mathbf{A}} \mid s', a^{\mathbf{H}}, a^{\mathbf{A}})$. We consider $\mathbf{H}$ policies $\pi^{\mathbf{H}}$ which, at timestep $t$, take as input the full history of $\mathbf{H}$'s observations and actions $h_t^{\mathbf{H}} \in (\Omega^{\mathbf{H}} \times \mathcal{A}^{\mathbf{H}})^t$ and map to a distribution over actions $\Delta \mathcal{A}^{\mathbf{H}}$. $\mathbf{A}$'s policy $\pi^{\mathbf{A}} : (\Omega^{\mathbf{A}} \times \mathcal{A}^{\mathbf{A}})^t \to \mathcal{A}^{\mathbf{A}}$ is analogous. We call $\pi^{\mathbf{H}}$ a best response to $\pi^{\mathbf{A}}$ when $\pi^{\mathbf{H}}$ maximizes expected discounted reward given $\pi^{\mathbf{A}}$, i.e., $\pi^{\mathbf{H}} \in \arg\max_{\hat{\pi}^{\mathbf{H}}} \mathbb{E}_{\hat{\pi}^{\mathbf{H}}, \pi^{\mathbf{A}}} \left[ \sum_{t=0}^{\infty} \gamma^t R(s_t, a_t^{\mathbf{H}}, a_t^{\mathbf{A}} \mid \theta) \right]$, where the expectation is taken over trajectories induced by the policies $(\pi^{\mathbf{H}}, \pi^{\mathbf{A}})$ and initial distribution $P_0$. The best response for $\mathbf{A}$ is defined analogously. A policy pair $(\pi^{\mathbf{H}}, \pi^{\mathbf{A}})$ is optimal if it maximizes the expected discounted reward in the POAG: $(\pi^{\mathbf{H}}, \pi^{\mathbf{A}}) = \arg\max_{\hat{\pi}^{\mathbf{H}}, \hat{\pi}^{\mathbf{A}}} \mathbb{E}_{\hat{\pi}^{\mathbf{H}}, \hat{\pi}^{\mathbf{A}}} \left[ \sum_{t=0}^{\infty} \gamma^t R(s_t, a_t^{\mathbf{H}}, a_t^{\mathbf{A}} \mid \theta) \right]$.

Note that optimal policy pairs are in particular Nash equilibria for the shared reward function $R$. Computationally, POAGs are equivalent to 2-player decentralized partially observable Markov decision processes (DecPOMDPs). Thus, finding optimal policy pairs for POAGs is NEXP-hard in general (Bernstein et al., 2002) (cf. Reif, 1984). A POAG may have multiple distinct optimal policy pairs, as there may be different ways for $\mathbf{H}$ and $\mathbf{A}$ to communicate or resolve coordination problems.

While the examples in this paper are simple, POAGs—and thus all our positive results—inherit the broad generality of DecPOMDPs. POAGs can model games where $\mathbf{H}$ acts first, where $\mathbf{A}$ acts first, or where $\mathbf{H}$ and $\mathbf{A}$ act simultaneously. POAGs allow both $\mathbf{H}$ and $\mathbf{A}$ to observe private information at multiple times, as well as take actions that influence both the state of the world and each other's observations.

### 2.2. Beliefs and Calibration of Beliefs

We are motivated to study observation interference because of its potential impact on $\mathbf{H}$'s belief about the state of the world. If $\mathbf{A}$ interferes with observations, could this cause $\mathbf{H}$ to have false beliefs?

To address this question, we apply known techniques to establish what information $\mathbf{H}$ needs to form calibrated beliefs in a POAG. (See Appendix A for proofs.) The simplest case of $\mathbf{H}$ knowing $\mathbf{A}$'s policy is when $\mathbf{A}$ is playing a fixed policy:

**Proposition 2.2.** *Suppose* $\mathbf{A}$ *is playing a fixed policy. If*

$\mathbf{H}$ *knows* $\mathbf{A}$'s *policy along with the POAG specification* $M$, *then* $\mathbf{H}$ *can form calibrated beliefs about the world state. For any timestep* $t$ *and state* $s_t$, $\mathbf{H}$ *can form* $P(s_t \mid o_{1:t}^{\mathbf{H}})$, *the probability of* $s_t$ *given* $\mathbf{H}$'s *observation history* $o_{1:t}^{\mathbf{H}}$.

In an iterated setting where $\mathbf{A}$ updates its policy between iterations, $\mathbf{H}$ can form beliefs if $\mathbf{H}$ additionally knows the policy update rule.

**Proposition 2.3.** *Suppose* $\mathbf{A}$ *is updating its policy each iteration of the game. Knowledge of the game dynamics, of* $\mathbf{A}$'s *initial policy, and of* $\mathbf{A}$'s *update rule is sufficient for* $\mathbf{H}$ *to form calibrated beliefs about* $\mathbf{A}$'s *future policy and of the world state.*

**Remark 2.4.** *Propositions 2.2 and 2.3 hold even if* $\mathbf{A}$ *is interfering with observations (Definition 3.2).*

**Remark 2.5.** *Proposition 2.2 and Proposition 2.3 continue to hold if* $\mathbf{H}$ *only knows a* prior *over* $\mathbf{A}$'s *policy.* $\mathbf{H}$ *can form a posterior using Bayes' rule; the posterior is calibrated if the prior is calibrated.*

When $\mathbf{H}$ knows $\mathbf{A}$'s policy, the preceding results show that $\mathbf{H}$ can form calibrated beliefs about the world, even when $\mathbf{A}$ is interfering with observations. *Observation interference increases* $\mathbf{H}$'s *uncertainty, but it doesn't break the calibration of* $\mathbf{H}$'s *beliefs.* Because $\mathbf{H}$ can still form calibrated beliefs in this setting, our work uses the concept of "interference" rather than the concept of "deception."

## 3. Defining Observation Interference

**Observation Interference**   First, we define what interference means. Intuitively, interference is taking action so that the human receives a less informative signal about the state. In particular, the human receives, in some sense, a *subset* of the information. We formalize this by saying one signal is less informative than another about the state if (without knowing the state) we could generate one signal from the other (cf. Blackwell et al., 1951; Blackwell, 1953; de Oliveira, 2018).

**Definition 3.1.** *Let* $(P(\cdot \mid s))_{s \in \mathcal{S}}$ *and* $(\hat{P}(\cdot \mid s))_{s \in \mathcal{S}}$ *be families of probability distributions over* $\Omega$. *We say that* $\hat{P}$ *is at most as informative as* $P$ *if there exists a stochastic function* $F : \Omega \rightsquigarrow \Omega$ *(mapping observations to random variables over observations) s.t. for all states* $s$ *we have* $F(X) \sim \hat{P}(\cdot \mid s)$ *if* $X \sim P(\cdot \mid s)$. *We say that* $P$ *is* (strictly) *more informative than* $\hat{P}$ *if* $P$ *is at least as informative as* $\hat{P}$ *but not vice versa.*

Why do we include the condition "for all states $s$" in Definition 3.1? Intuitively, we want it always to be possible to use the stochastic function $F$ to reconstruct the less informative signal from the more informative signal. Since our setting is partially observable, the "for all states $s$" condition allows a

player to do this reconstruction in any scenario, even if their observations don't enable them to infer the state.

With this definition in hand, we define an observation-interfering action as one that results in the human's observation being less informative about the state than the observation distribution resulting from another assistant action. We additionally require that this other action has the same effects on the state and immediate reward. After all, it is clear that sometimes **A** has to trade off providing information to **H** with optimizing its effect on the environment.

**Definition 3.2.** *Let $M$ be any POAG. We say that $\hat{a}^{\mathbf{A}}$ is observation-interfering if there exists some other action $a^{\mathbf{A}}$ s.t. $\hat{a}^{\mathbf{A}}$ and $a^{\mathbf{A}}$ have the same effect on state transitions and immediate rewards, but for all $a^{\mathbf{H}}$, we have that $(O^{\mathbf{H}}(\cdot \mid a^{\mathbf{H}}, s, a^{\mathbf{A}}))_{s \in S}$ is more informative than $(O^{\mathbf{H}}(\cdot \mid a^{\mathbf{H}}, s, \hat{a}^{\mathbf{A}}))_{s \in S}$.*

To discuss policies that play observation-interfering actions, we use the following definition:

**Definition 3.3.** *We say that a policy $\pi^{\mathbf{A}}$ interferes with observations at the action level (or equivalently, takes observation-interfering actions) in a POAG $M$ if there is any history $h \in (\Omega^{\mathbf{A}} \times \mathcal{A}^{\mathbf{A}})^*$ where $\pi^{\mathbf{A}}(\cdot \mid h)$ assigns positive probability to an observation-interfering action.*

**Lack of Private Information**    To understand the conditions under which interference occurs, it is useful to consider POAGs where one of the players has no private information.

**Definition 3.4.** *For a POAG $M$, we say $\mathbf{A}$ has no private information if there exists a function $f$ determining $\mathbf{A}$'s observations from $\mathbf{H}$'s observations. For all state-action tuples $(s', a^{\mathbf{H}}, a^{\mathbf{A}})$ and observation pairs $(o^{\mathbf{H}}, o^{\mathbf{A}}) \in \text{supp}(O(\cdot, \cdot \mid s', a^{\mathbf{H}}, a^{\mathbf{A}}))$, then $f$ must have $f(o^{\mathbf{H}}) = o^{\mathbf{A}}$.*

**Communication**    To further understand the motivations behind interference, we will also consider POAGs in which the players are able to directly communicate. Thus, for any given POAG, the following defines a variant of that POAG in which the players have an additional channel for communication. We will always assume that the channel has enough bandwidth for the sender to share all private information, i.e., that there is an injection from the sender's observation space into the message space.

**Definition 3.5.** *Let $M$ be a POAG. Define $M^{\mathbf{A} \to \mathbf{H}}$, $M^{\mathbf{H} \to \mathbf{A}}$, and $M^{\mathbf{H} \leftrightarrow \mathbf{A}}$ as a variants of $M$ with unbounded communication channels. We define $M^{\mathbf{H} \to \mathbf{A}}$ below; $M^{\mathbf{A} \to \mathbf{H}}$ and $M^{\mathbf{H} \leftrightarrow \mathbf{A}}$ are analogous.*

*To construct $M^{\mathbf{H} \to \mathbf{A}}$, let $\mathcal{M}$ be some set of possible messages/signals s.t. there is an injection $\Omega^{\mathbf{A}} \hookrightarrow \mathcal{M}$. Then, construct a new human action space $\hat{\mathcal{A}}^{\mathbf{H}} = \mathcal{A}^{\mathbf{H}} \times \mathcal{M}$ and new assistant observation space $\hat{\Omega}^{\mathbf{A}} = \Omega^{\mathbf{A}} \times \mathcal{M}$. The new observation kernel has $\hat{O}\left(o^{\mathbf{H}}, (o^{\mathbf{A}}, m') \mid s', (a^{\mathbf{H}}, m), a^{\mathbf{A}}\right) =$*

$\mathbb{1}[m{=}m']O(o^{\mathbf{H}}, o^{\mathbf{A}} \mid s', a^{\mathbf{H}}, a^{\mathbf{A}})$. *For everything else, the messages are simply ignored.*

**Plausible Human Policies**    We may have various expectations on how **H** will play in a POAG. Especially if there are multiple optimal policy pairs, we may expect some of these policy pairs to be more plausible because they require simpler behavior of the human (cf. Hu et al., 2020; Treutlein et al., 2021). Both of the conditions below are based on the idea that **A** and **H** are unlikely to use consequential actions in the world to communicate with each other.

Our first condition intends to express a form of naivete on **H**'s part in how she interprets her observations. Roughly, the condition says that **H** takes her observations at face value, i.e., as if they were not interfered with. She does not try to interpret them as a form of communication by **A**. For instance, if **H** reads a thermometer as saying that a temperature is 37 degrees, she chooses under the assumption that the temperature is indeed 37 degrees, rather than, say, interpreting 37 as a message sent by **A**.

**Definition 3.6.** *We say that a human policy $\pi^{\mathbf{H}}$ observes naively if $\pi^{\mathbf{H}}$ is a best response to some $\pi^{\hat{\mathbf{A}}}$ that does not interfere with observations at the action level.*

The second property is that when the human knows that her action has no effect on the state, then she chooses among actions that maximize immediate reward. To state this formally, we first define the following. We say that in $h_t^{\mathbf{H}}$ *actions don't affect state transitions*, if for all $s$ s.t. we have $P(s \mid h_t^H, \pi^{\mathbf{A}}) > 0$ for some $\pi^{\mathbf{A}}$, we have that for all $a^{\mathbf{A}}$ the transition probability $P(s' \mid s, a^{\mathbf{A}}, a^{\mathbf{H}})$ is constant over $a^{\mathbf{H}}$. We say that $\pi^{\mathbf{H}}$ *myopically maximizes reward* in $h_t^{\mathbf{H}}$ if there is some distribution $\alpha^{\mathbf{A}} \in \Delta(\mathcal{A}^{\mathbf{A}})$ s.t. $\pi^{\mathbf{H}}(\cdot \mid h_t^{\mathbf{H}})$ randomizes only over actions in $\arg\max_{a^{\mathbf{H}}} \mathbb{E}_{a^{\mathbf{A}} \sim \alpha^{\mathbf{A}}, s \sim P(\cdot \mid h_t^{\mathbf{H}}, a^{\mathbf{H}}, a^{\mathbf{A}})} \left[ R(s, a^{\mathbf{H}}, a^{\mathbf{A}}, \theta) \right]$. (Intuitively, $\alpha^{\mathbf{A}}$ is **H**'s belief about what action **A** is going to take.)

**Definition 3.7.** *We say that a human policy $\pi^{\mathbf{H}}$ acts naively if whenever **H** faces a choice that doesn't affect state transitions (but potentially affects **A**'s observation), **H** plays an action that myopically maximizes reward.*

Importantly, if **H** acts naively, she is unwilling to play a suboptimal action to communicate information to **A**.

# 4. Communicating Private Information is an Incentive for Observation Interference

## 4.1. Revealing Errors can Emerge as an Optimal POAG Solution

We can model RLHF within the POAG framework as follows: **A**'s goal in RLHF is to satisfy **H**'s preferences. In a POAG, this corresponds to the shared reward function $R$

which has a parameterization $\theta$ that only **H** knows. In RLHF, **A** rolls out trajectories, and **H** picks which trajectory is preferred. A POAG can model this by letting **H** observe pairs of trajectories explored by **A** but only giving **H** a binary action (to choose which trajectory **H** prefers). In RLHF, **A**'s final policy maximizes an estimate of $R$ based on a dataset of **H**'s preference comparisons (Lang et al., 2024, Proposition 4.1). In the POAG framework, **A** can compute this policy based on **A**'s observations of **H**'s binary actions.

Past work has shown how RLHF can cause misleading (Wen et al., 2024) and deceptive (Williams et al., 2024; Lang et al., 2024) behaviors. Specifically, Lang et al. (2024, Example B.1) show that in order to get better human feedback, RLHF can have an incentive to *hide error messages*.

In contrast to RLHF, we show with the following example that *revealing error messages* can emerge in POAG solutions.

**Example 4.1.** *First,* **A** *is executing on a remote machine where logging has been disabled by default.* **A** *takes one of two actions: (1) Attempt to install* `cuda`. *The installation succeeds with 50% probability. An empty observation is produced (since logging is disabled). (2) Re-enable logging and attempt to install* `cuda`. *The installation succeeds with 50% probability. An observation is produced containing a success or failure message.*

*Then,* **H** *takes one of two actions: (1) Run an experiment. If* `cuda` *is installed successfully, this yields +1 reward. Otherwise, it yields -2 reward. (2) Don't run an experiment. This always yields 0 reward.*

*In the optimal policy pair,* **A** *reenables logging; this reveals errors to* **H**!

In fact, if **A** has no private information, then it never needs to take observation-interfering actions for an optimal solution!

**Theorem 4.2.** *Let $M$ be any POAG. Let* **A** *have no private information. Then there is an optimal policy pair $(\pi^{\mathbf{H}}, \pi^{\mathbf{A}})$ for $M$ in which $\pi^{\mathbf{A}}$ does not interfere with observations at the action level (and $\pi^{\mathbf{H}}$ observes naively).*

## 4.2. Communicating Private Information is an Incentive for Observation Interference at the Action Level

One might hope that **A** would never take observation-interfering actions. After all, classic theory tells us that when **H** is in a single-agent setting, the value of (perfect) information is nonnegative (e.g., Koller & Friedman, 2009, Sect. 23.7; Russell & Norvig, 2010, Sect. 16.6.3): more informative observations never lead to worse solutions. But as it turns out, when **H** and **A** interact, there are some POAGs in which all optimal policy pairs require **A** to take observation-interfering actions. The main reason for **A** to take observation-interfering actions is to communicate its own private information to **H**. Consider the following example.

**Example 4.3.** **H** *has typed* `apt list -a cuda` *to see the list of* `cuda` *versions available to be installed. Out of 10 total versions, only a (non-empty) subset are available. And of these available versions, only a subset are compatible with the other environment software.*

*First,* **A** *takes an action. For each of the 10 total* `cuda` *versions,* **A** *can choose to or not to suppress it from the list of available packages. This gives* **A** $2^{10}$ *total actions, where 1 action is non-observation interference (suppressing nothing), and the remaining $2^{10} - 1$ actions interfere with observations.*

*Second,* **H** *takes an action.* **H** *has 10 possible actions which try to install the corresponding version of* `cuda` *if it appears in the version list. If an available* `cuda` *version that is compatible with the other environment software is installed, it yields +1 reward. Otherwise, it yields 0 reward.*

*Suppose* **A** *sees which versions are compatible with the other software in the environment, but* **H** *doesn't. Then* **A**'s *optimal policy is to suppress the versions of* `cuda` *that are incompatible.*

Our high-level takeaway from this example is that in some POAGs, all optimal policy pairs require **A** to take observation-interfering actions. Importantly, in the optimal policy pair for the above example, **H** observes naively. In particular, the above doesn't require **H** and **A** to have some communication protocol and for **H** to interpret her observations as encoding **A**'s beliefs. **H** can act as if no interference is happening. We thus summarize the high-level takeaways in the following result, with details in Appendix B.3.

**Proposition 4.4.** *There exists a POAG $M$ where all optimal policy pairs $(\pi^{\mathbf{A}}, \pi^{\mathbf{H}})$ have that $\pi^{\mathbf{A}}$ interferes with observations at the action level and that $\pi^{\mathbf{H}}$ observes and acts naively.*

Intuitively, in Example 4.3, **A** interferes in order to convey information to **H**. **A** knows **H**'s optimal choice, but cannot tell her. So, **A** needs to interfere in a way that leads **H** to the optimal choice.

The need for **A** to take observation-interfering actions to communicate to **H** disappears if **A** has other means of communication. For instance, if in Example 4.3, **A** could simply tell **H** what to do, then **A** wouldn't need to interfere. To formalize this intuition, we now prove that if **A** can communicate with **H**, then there is always an optimal policy pair that does not require interference.

**Theorem 4.5.** *Let $M$ be any POAG, and provide* **A** *with an unbounded communication channel to* **H***, forming $M^{\mathbf{A} \rightarrow \mathbf{H}}$. Then there is an optimal policy pair $(\pi^{\mathbf{H}}, \pi^{\mathbf{A}})$ for $M^{\mathbf{A} \rightarrow \mathbf{H}}$ where $\pi^{\mathbf{A}}$ does not interfere with observations at the action*

*level and $\pi^{\mathbf{H}}$ observes naively.*

One could argue that in practice, an unrestricted communication channel between $\mathbf{A}$ and $\mathbf{H}$ could usually be made available. However, Theorem 4.5 ignores various real-world obstacles. For one, it considers communication that incurs no cost, but in reality, communication costs $\mathbf{H}$ time and effort. Second, the optimal policy pair requires $\mathbf{A}$ to send information in a way that $\mathbf{H}$ can reliably understand and act upon. We expect that in practice, $\mathbf{A}$ and $\mathbf{H}$ sometimes cannot understand each other. Therefore, despite Theorem 4.5, we think observation interference is of broad practical relevance, even where $\mathbf{A}$ can, e.g., send text messages to $\mathbf{H}$.

### 4.3. Optimal Policy Pairs Never Require Observation Interference at the Policy Level

In Definition 3.2, we first define observation interference as a feature of actions. We then say in Definition 3.3 that a policy interferes with observations at the action level if and only if it ever takes an observation-interfering action.

Because the definition is ultimately about actions, it doesn't consider how $\pi^{\mathbf{A}}$ might choose to take observation-interfering actions in a way that depends on $\mathbf{A}$'s observations. To account for $\pi^{\mathbf{A}}$'s dependence on its observation, we define an alternative notion of what it means for a policy to interfere with observations.

Let $P_{o_t^{\mathbf{H}}}$ be the distribution over human observations at time $t$. Further, let $L_t(\pi^{\mathbf{H}}, \pi^{\mathbf{A}})$ be the set of possible states at time $t$.

**Definition 4.6.** *Let $M$ be a POAG. We say that $\mathbf{A}$'s policy $\hat{\pi}^{\mathbf{A}}$ interferes with observations at the policy level if there exists some other partial policy $\pi_t^{\mathbf{A}}$ for time step $t$ s.t. $\hat{\pi}_t^{\mathbf{A}}$ and $\pi_t^{\mathbf{A}}$ have the same effect on state transitions and immediate rewards, but for all $\pi^{\mathbf{H}}$ we have that $P_{o_{t+1}^{\mathbf{H}}}(\cdot \mid \pi^{\mathbf{H}}, s_{t+1}, \hat{\pi}_{0:t}^{\mathbf{A}}, \pi^{\mathbf{H}})_{s_{t+1} \in L_{t+1}(\pi^{\mathbf{H}}, \hat{\pi}_{0:t}^{\mathbf{A}})}$ is less informative than the corresponding distribution if we replace $\hat{\pi}_{0:t}^{\mathbf{A}}$ with $(\hat{\pi}_{0:t-1}^{\mathbf{A}}, \pi_t^{\mathbf{A}})$.*

Compared to our previous action-level notion of observation interference (Definition 3.2), this new policy-level notion (Definition 4.6) differs in how it treats $\mathbf{H}$'s inference process. Whereas the action-level notion models inference about isolated observations, the policy-level notion allows $\mathbf{H}$ to make inferences in the context of $\mathbf{A}$'s overall strategy.

We now revisit Example 4.3. For observation tampering under our earlier Definition 3.2, $\mathbf{H}$ simply knows that $\mathbf{A}$ has taken the *action* to suppress some versions of cuda; $\mathbf{H}$ does not know anything about $\mathbf{A}$'s *policy*. For all $\mathbf{H}$ knows, $\mathbf{A}$'s policy could be to randomly suppress cuda versions or to always suppress the same cuda version. Thus, suppressing any version is strictly less informative for $\mathbf{H}$ than the list of all available versions. This is why Definition 3.2 calls

suppressing versions "tampering at the action level."

The key difference with Definition 4.6 is that $\mathbf{H}$ knows $\mathbf{A}$'s policy. Suppose that $\mathbf{A}$'s policy $\pi^{\mathbf{A}}$ is to suppress exactly the versions of cuda that are incompatible with the other software in the environment. Because $\mathbf{H}$ knows that $\mathbf{A}$ suppressed the incompatible cuda versions, seeing the filtered list tells $\mathbf{H}$ which versions of cuda are compatible! Although suppressing versions is strictly less informative under Definition 3.2 (when $\mathbf{H}$ doesn't know $\mathbf{A}$'s policy), suppressing versions provides $\mathbf{H}$ with new information under Definition 4.6 (when $\mathbf{H}$ knows $\mathbf{A}$'s policy). Accordingly, $\pi^{\mathbf{A}}$ is interfering with observations at the action level *but not at the policy level*.

In our examples, we will mostly consider actions that in some sense act directly on $\mathbf{H}$'s observations. Yet Definition 4.6 also considers the informational effects of physical actions. For example, if $\mathbf{A}$ (visibly) tries to open a door that $\mathbf{A}$ knows to be locked, then this reveals to $\mathbf{H}$ that the door is locked. Consequently, not trying to open the door (when $\mathbf{A}$ knows it to be locked) is an instance of observation interference in the sense of Definition 4.6. While having the same (null) effect on the state of the world, trying to open the door provides $\mathbf{H}$ with more information.

As in Example 4.3, cases which appear to destroy information when viewed at the action level may actually provide new information when viewed at the policy level. In fact, the following theorem shows that it's *never* strictly necessary to interfere with observations at the policy level.

**Theorem 4.7.** *Let $M$ be any POAG. Then there exists an optimal policy pair $(\pi^{\mathbf{H}}, \pi^{\mathbf{A}})$ for $M$ s.t. $\pi^{\mathbf{A}}$ does not interfere with observations at the policy level.*

This contrasts with Proposition 4.4: whereas it is sometimes necessary to interfere with observations at the *action* level, it is never necessary at the *policy* level.

The main idea behind this proof is similar to the proof of Theorem 4.2 (given in Appendix B.3). That is, if we start with an optimal policy in which $\mathbf{A}$ observation-interferes, then we can replace $\mathbf{A}$'s policy with the corresponding more informative policy and update $\mathbf{H}$'s policy to imitate the garbling. The proof of Theorem 4.2 considers the set of *actions*, which is finite. The main extra difficulty in proving Theorem 4.7 is that we must deal with spaces of *policies*, which may be infinitely large. Thus, if we replace a policy with a more informative one, there might be a new policy which is even more informative, and so on forever.

Note that there are many possible ways to extend or refine Definitions 3.2 and 4.6 in ways that preserve our key results. We choose Definitions 3.2 and 4.6 in part for their simplicity; for more discussion of this point, see Appendix G.

## 5. Querying H's Preferences is an Incentive for Observation Interference

We now study a second reason $\mathbf{A}$ can have for interfering with observations. We have already shown (Theorems 4.2, 4.5 and 4.7) that even if $\mathbf{H}$ has private information and no communication channel, there's always an optimal policy pair in which $\mathbf{A}$ does not interfere, as long as $\mathbf{A}$ doesn't have private information. So, if $\mathbf{H}$ plays a best response to $\mathbf{A}$'s policy, then $\mathbf{A}$ can choose a non-interference policy without loss of utility. However, if $\mathbf{H}$ does not play a best response to $\mathbf{A}$, then reasons for interference emerge that are more subtle than those in the $\mathbf{A} \rightarrow \mathbf{H}$ case.

Intuitively, $\mathbf{A}$ might need to interfere with observations to elicit $\mathbf{H} \rightarrow \mathbf{A}$ communication. Suppose $\mathbf{A}$ needs some information from $\mathbf{H}$, but $\mathbf{H}$ is acting naively (see Definition 3.7) in a way that does not reveal her private information. By changing $\mathbf{H}$'s observation, $\mathbf{A}$ can make $\mathbf{H}$'s naive response communicate useful information to $\mathbf{A}$. The following example illustrates this phenomenon.

**Example 5.1.** $\mathbf{H}$ *would like to schedule a job on a cluster. She can choose between two nodes. By default, she receives a signal from the environment about the two nodes' specifications. Each node may be either GPU-optimized or CPU-optimized. Also, the CPUs may be either AMD or Intel.*

$\mathbf{H}$ *has a strong preference between GPU-optimized and CPU-optimized nodes. She has a weak preference between AMD and Intel. These preferences are unknown to* $\mathbf{A}$.

$\mathbf{A}$ *can interfere with* $\mathbf{H}$*'s observation about the available nodes. In particular,* $\mathbf{A}$ *can make it so that a choice between two CPU-optimized nodes appears as a choice between a GPU-optimized and CPU-optimized node.* $\mathbf{A}$ *observes* $\mathbf{H}$*'s choice. Later,* $\mathbf{A}$ *is charged with scheduling a job for* $\mathbf{H}$ *and has to choose between a CPU- and a GPU-optimized node on* $\mathbf{H}$*'s behalf.*

*If* $\mathbf{H}$ *chooses naively upon seeing only CPU-optimized nodes (simply choosing her favorite), then* $\mathbf{A}$*'s best response interferes with observations at both the action and policy levels. Interfering with observations allows* $\mathbf{A}$ *to learn* $\mathbf{H}$*'s preference about GPU- vs CPU-optimized nodes.*

In Example 5.1, one might ask why $\mathbf{A}$ can't just ask $\mathbf{H}$ each time $\mathbf{A}$ makes a decision. Simply asking $\mathbf{H}$'s preference is reasonable when $\mathbf{A}$ has only one decision to make. However, we are motivated by cases where $\mathbf{A}$ has many decisions to make, and asking $\mathbf{H}$'s preferences each time would be cumbersome.

At first sight, Example 5.1 may appear to be a counterexample to Theorem 4.2, which states that a non-interfering optimal policy pairs exists. However, note that Example 5.1 actually *does* have optimal policy pairs in which $\mathbf{A}$ doesn't

interfere. In particular, even if $\mathbf{A}$ does not interfere and the two available nodes are CPU-optimized, $\mathbf{H}$ may simply communicate her CPU-versus-GPU preference anyway! That is, when facing a choice between CPU-optimized node 1 and 2, she may choose, say, 1 if she favors GPU-optimized nodes and 2 if she favors CPU-optimized nodes. However, this type of human strategy seems implausible, as it would require $\mathbf{H}$ and $\mathbf{A}$ to have settled on some communication strategy that overrides $\mathbf{H}$'s immediate preferences about the machines that $\mathbf{H}$ can in fact choose between.

The key point of Example 5.1 is that—while there is *some* optimal policy pair without observation interference—there is no *plausible* optimal policy pair that avoids observation interference. More specifically, we use the notion of acting naively (Definition 3.7) to express this notion of plausibility and rule out the above policy. We thus obtain the following proposition (with proof in Appendix C): in some POAGs, if we want to play an optimal policy pair and we want $\mathbf{H}$ to be able to act naively, then $\mathbf{A}$ has to interfere with observations.

**Proposition 5.2.** *There is a POAG $M$ with the following properties. For every optimal policy pair $(\pi^{\mathbf{H}}, \pi^{\mathbf{A}})$, at least one of the following holds: (i) $\pi^{\mathbf{H}}$ is not acting naively, or (ii) $\pi^{\mathbf{A}}$ interferes with observations at both the action and policy levels. Additionally, there exists an optimal policy pair $(\pi^{\mathbf{H}}, \pi^{\mathbf{A}})$ where $\pi^{\mathbf{H}}$ acts naively and $\pi^{\mathbf{A}}$ interferes with observations at both the action and policy levels.*

*These properties continue to hold if we require that in $M$, $\mathbf{A}$ has no private information or can arbitrarily send messages to $\mathbf{H}$ (i.e., there is a POAG $\tilde{M}$ s.t. $M = \tilde{M}^{\mathbf{A} \rightarrow \mathbf{H}}$).*

Intuitively, the problem in the above example is that the human has private information that she needs to communicate with her choices. (Because her choices yield different immediate rewards, naive choices fail to communicate.) As before, the need for interference or non-naive choice disappears if the human has no private information to provide. The following shows that the need for interference / non-naivete also disappears if $\mathbf{H}$ can communicate with $\mathbf{A}$. To also rule out the need to interfere with observations for $\mathbf{A} \rightarrow \mathbf{H}$ communication (discussed in Section 4.2) we assume communication channels in both direction.

**Theorem 5.3.** *Let $M$ be a POAG. There exists an optimal policy pair $(\pi^{\mathbf{H}}, \pi^{\mathbf{A}})$ for $M^{\mathbf{H} \leftrightarrow \mathbf{A}}$ where $\pi^{\mathbf{H}}$ is naive and assumes honesty while $\pi^{\mathbf{A}}$ does not interfere at either the action or policy levels.*

## 6. Human Irrationality is an Incentive for Observation Interference

Finally, we consider a third reason for observation interference: human irrationality or bounded rationality. Roughly, reducing the amount of information supplied to the human may simplify the human's decision problem and thus im-

prove her decision making. Importantly, this motivation for observation interference may exist even if neither **H** nor **A** has any private information.

As our model of human decision making, we adopt Boltzmann rationality (Luce, 1959; McFadden, 1973), which has recently been used in (C)IRL (Laidlaw & Dragan, 2021; Ramachandran & Amir, 2007; Ziebart et al., 2008).

**Definition 6.1.** *Let $M$ be a POAG. Let $\pi^{\mathbf{A}}$ be **A**'s policy in $M$. We say that **H**'s policy $\pi^{\mathbf{H}}$ is a Boltzmann-rational response to $\pi^{\mathbf{A}}$ if there exists some $\beta > 0$ s.t. for every human observation history $h$ that arises with positive probability in $M$ under $(\pi^{\mathbf{A}}, \pi^{\mathbf{H}})$ we have that $\pi^{\mathbf{H}}(a \mid h) \propto \exp\left( \beta \mathbb{E}\left[ \sum_{t'=t}^{\infty} \gamma^{t'} R(S_t, A_t^{\mathbf{A}}, A_t^{\mathbf{H}}) \mid \pi^{\mathbf{H}}, \pi^{\mathbf{A}}, h \right] \right)$.*

It turns out that even if the Boltzmann-rational human has calibrated beliefs, **A**'s optimal policy sometimes interferes with observations, even if neither **A** nor **H** has private information. Intuitively, providing more information may sometimes result in less clear-cut decisions, i.e., decision situations with a smaller difference between the correct and incorrect option. To illustrate this phenomenon, consider the following example.

**Example 6.2.** *H is running a terminal command and is unsure whether to run the command with flag 1 or flag 2. With equal probability, either flag 1 or flag 2 is better, and how good the flags are differs by either a little or a lot. The worse flag always yields a utility of 0, while the better flag either yields a utility of 1 or a utility 7. Thus, **H** is uniformly at random in one of four states. **A** has two actions:* `man` *and* `tldr`*. The* `man` *page is a long document that tells the human exactly what the values of the flags are (i.e., the exact state: which flag is better and whether its utility is 1 or 7). The* `tldr` *page is a short summary that tells the human which flag is better, but not by how much (i.e., ruling out half the states, leaving half remaining). Thus, the expected utility of the better flag is 4 (and of the worse flag is 0).*

Intuitively, both the `tldr` and `man` pages allow the human to choose optimally, but the `man` page is more complicated and therefore more likely to be misinterpreted. Choosing specific utilities, the effect of interference under Boltzmann rationality is as follows. If **A** interferes (i.e., provides the `tldr` page), then **H** always chooses between a utility of 4 and 0. If **A** does not interfere, then half the time, **H** chooses between utilities 1 and 0, and half the time **H** chooses between utilities 7 and 0. It turns out that for $\beta = 1$, **H** achieves higher utility in expectation under the condition where **A** interferes. Building on this idea, we can prove the following (with details in Appendix D).

**Proposition 6.3.** *For every $\beta > 0$, $\exists$ a POAG in which neither **H** nor **A** has private information s.t. all $\beta$-Boltzmann-rational/optimal policy pairs $(\pi^{\mathbf{H}}, \pi^{\mathbf{A}})$ have $\pi^{\mathbf{A}}$ interfere with observations at both the action and policy levels.*

# 7. Experiments

Motivated by the theory in Section 4 and Section 6, we develop a model game and run experiments to analyze:

1. How does the amount of **H**'s irrationality affect **A**'s incentive to take observation-interfering actions?

2. How does the amount of **A**'s private information affect **A**'s incentive to take observation-interfering actions?

## 7.1. Experiment Details

We study a game where selecting the best action requires combining private observations known only to **H** and private observations known only to **A**. The game presents **A** with a tradeoff: **A** can interfere with observations to communicate information that only **A** observes, but interfering also destroys information that only **H** observes.

Concretely, the game has $d$ products. Each product $i$ has two attributes, $H_i$ and $R_i$, drawn i.i.d. from Unif$(0, 1)$. Each product's utility is the sum of its attributes, $U_i = H_i + R_i$. The game consists of two moves. First, **A** sees $R_i$ for $i = 1, \ldots, k$ where $k$ is the number of **A**'s private observations. **A** chooses a set of products to interfere with. For the products **A** interfered with, **H** sees $\hat{H}_i = -\infty$; for the remaining products, **H** sees $\hat{H}_i = H_i$. Second, **H** chooses a product $a_i$. Both **H** and **A** receive a common payoff of the chosen product's utility, $U_i$.

We assume the human's product selection policy is Boltzmann rational over their observed values $\hat{H}_i$:

**Definition 7.1.** *H's Boltzmann selection policy chooses products by a Boltzmann distribution over $\hat{H}_i$, the observed product values: $\pi^{\mathbf{H}}(a_i) \propto \exp(\beta \hat{H}_i)$. The parameter $\beta$ controls **H**'s rationality.*

We consider **A** policies that always interfere with $k$ observations for some fixed $k$. Call these policies $k$-interference. We study the optimal such policies, characterized by the following result:

**Proposition 7.2.** *Consider **A** policies that always interfere with $k$ observations for some fixed $k$. Among the $k$-interference policies for a given $k$, **A**'s best response to **H**'s straightforward product selection policy is as follows. **A** interferes with the $k$ smallest $\hat{R}_i$ values where $\hat{R}_i = R_i$ if **A** observes $R_i$, and $\hat{R}_i = 0.5$ otherwise.*

We consider a game with $d = 5$ products. We vary $R$'s number of interferences $k \in \{0, 1, 2, 3, 4\}$. We run a Monte Carlo simulation with 30,000 trials to calculate the expected payoff in each setting.

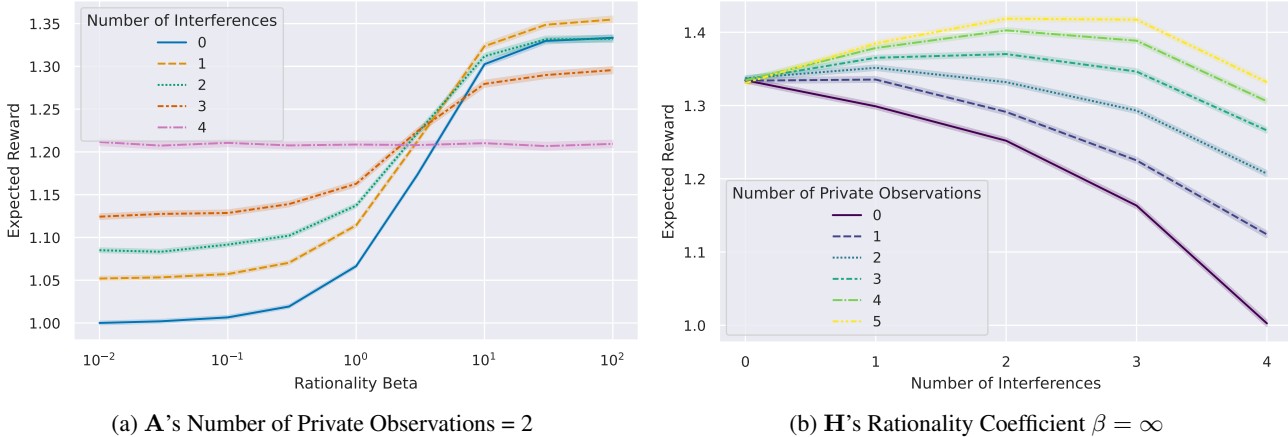

(a) **A**'s Number of Private Observations = 2        (b) **H**'s Rationality Coefficient $\beta = \infty$

Figure 1: Incentives to interfere with observations in the product selection game. (Left) When **H** is highly irrational, it's best for **A** to interfere, effectively making the choice for **H**. As **H** becomes more rational, there is an increasing cost to interference, and there's a tradeoff: **A** should interfere to communicate some information, but not destroy too much information by excessive interference. (Right) In line with Theorem 4.2, **A** has no incentive to interfere when **A** has no private observations. With more private observations, **A** has more incentive to interfere.

## 7.2. Varying H's Rationality

How does **H**'s rationality impact **A**'s incentive for observation interference? We fix **A** to have 2 private observations. We do a logarithmic sweep over **H**'s rationality coefficient $\beta \in \{0.01, 0.03, 0.1, 0.3, 1, 3, 10, 30, 100\}$. Figure 1a shows how the expected reward changes w.r.t. $\beta$.

When **H** is highly irrational at $\beta = 0.01$, **A** should interfere with as many sensors as possible, effectively choosing **H**'s action. When **H** is acting little better than randomly, it's best for **A** to choose **H**'s action, even when **A** has less information than **H**. For larger values of $\beta$, a tradeoff emerges. As **A** has two private observations, there is an increasing benefit to interfere to communicate information to **H**. However, as **H** can now make use of their own private observations, **A** must be careful not to destroy too much of **H**'s private information by excessive interference.

## 7.3. Varying A's Private Information

How does the amount of private information available to **A** influence **A**'s incentive for observation interference? In Theorem 4.2, we showed conditions under which private observations for **A** are a necessary condition for observation interference to occur. Now, we analyze the *degree* to which private observations incentivize observation interference. Based on Theorem 4.2, we hypothesize that there are circumstances where *more* private information leads to *more* observation interference.

We vary $R$'s number of private observations in $\{0, 1, 2, 3, 4, 5\}$. We consider **A**'s $k$-interference policies and analyze how the relative performance of different levels

of observation interference $k$ change with the number of private observations available to **A**.

Figure 1b shows how the expected reward changes depending on $k$, the number of interferences. When **A** has no private observations, then reward decreases for each increased number of interferences. However, as the number of **A**'s private observations increases, the relative ordering of the observation interference policies changes; *with more private observations,* **A** *has an incentive to interfere with more observations*. This confirms our hypothesis based on Theorem 4.2. Nevertheless, there is a limit to **A**'s observation interference incentive. Because interfering with observations destroys **H**'s information, **A** must be careful not to interfere too much.

## 8. Conclusion

**Limitations and Future Work** Optimal policy pairs sometimes require **H** and **A** to have a shared communication protocol (e.g., Example 5.1). It would be interesting to study additional solution concepts, such as correlated equilibria and communication equilibria, to handle this sort of communication (Forges, 1986). While we consider only a single human and single assistant, it would also be interesting to study scenarios with multiple humans and multiple assistants. As we focus on the Boltzmann rationality model of human decision making (Definition 6.1), future work could consider other human models and empirical validation with human subjects. Lastly, while we run experiments in one model of a POAG, it would be interesting to see if and how our experimental trends generalize to other POAGs, including POAGs where **A** must query **H**'s preferences.

## Impact Statement

AI assistants are being developed and deployed in settings where humans can only partially observe what's happening. For example, AI assistants including ChatGPT, Claude, and Gemini can search the web while only returning summaries to users (OpenAI, 2024; Anthropic, 2025; Google, 2024). Moreover, the sorts of AI models powering these assistants are processing increasingly long inputs. Whereas the original ChatGPT model could only process 4096 input tokens, today's Gemini 1.5 Pro can process 2,000,000 input tokens—which is roughly 100,000 lines of code, or 16 novels of average length in English (Google, 2025). In the future, we anticipate that AI assistants will be deployed at increasing scale, independently taking more actions on behalf of users and processing increasingly long context lengths. We thus expect that over time, humans will have less and less ability to directly observe everything that's happening.

Even when the AI assistant and the human have perfect value alignment, we show how observation interference can emerge from several distinct incentives. As we focus on optimal assistants—analyzing optimal policy pairs and best responses—all of the incentives for observation interference that we consider are done for the human's benefit. This creates a nuanced picture, suggesting that not all observation interference is inherently bad. As AI assistants might exhibit observation interference for a mix of good and bad reasons, it would be interesting for future work to explore how to handle this nuanced situation. For example, future AI systems could be designed with transparency about when interference occurs and user controls to override interference when desired.

With this theory, our goal is to understand the causes of observation interference and help disentangle them in practice. We intend for our work to help AI developers build assistants that their users can trust. Our work is primarily theoretical, and we are not aware of any ways it could be used to cause harm.

**Computational Complexity**  Given that finding optimal policies in POAGs is NEXP-hard, how might our results apply in a given environment? Most of our paper is descriptive, characterizing when observation tampering could happen. Complexity considerations could affect these results in either direction. It's easy to construct environments where finding good observation-interfering policies is computationally intractable but constructing good non-interfering policies is easy; and vice versa. In practice, complexities of the environment can be orthogonal to incentives to interfere. For instance, a real-world version of the CUDA example is complex (A assesses complicated software compatibility issues), but the decision whether to interfere with observa-

tions is easy. We believe our characterizations remain useful even in complex environments (where we can't expect optimal policies), although we can't make as definitive claims as we can about optimal policies.

We have discussed allowing communication between H and A. A complexity-theoretic argument favors this solution: If H and A share all private information, the game effectively turns from a DecPOMDP into a POMDP. Solving POMDPs is PSPACE-complete and thus likely easier than solving DecPOMDPs.

## Author Contributions

The project was conceived by S.E., who developed the initial theorems and examples. C.O. led the theoretical development, proving the main results and formalizing the examples. S.E. conducted the experimental analysis. The manuscript was written collaboratively, with S.E. leading the introduction, experimental, and conclusion sections, while the theoretical sections were co-written by S.E. and C.O. V.C. and S.R. advised throughout the project.

## Acknowledgments

The authors thank Mark Bedaywi, Emery Cooper, Anca Dragan, Andrew Garber, Linus Luu, and Rohan Subramani for helpful discussions. C.O.'s work is supported by an FLI AI Existential Risk Fellowship. V.C. acknowledges financial support from the Cooperative AI Foundation, Polaris Ventures (formerly the Center for Emerging Risk Research), and Jaan Tallinn's donor-advised fund at Founders Pledge. S.E. and S.R. are grateful for Open Philanthropy's gift to the Center for Human-Compatible AI.

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

## A. Proofs for Section 2.2

Our techniques are similar to those of Shah et al. (2020) and Desai (2017), who show how to form a single-agent POMDP for **A** by embedding **H** into the environment dynamics. However, our construction works in the opposite direction, with **H** embedding **A**'s actions and observations into the environment.

**Proposition 2.2.** *Suppose **A** is playing a fixed policy. If **H** knows **A**'s policy along with the POAG specification $M$, then **H** can form calibrated beliefs about the world state. For any timestep $t$ and state $s_t$, **H** can form $P(s_t \mid o_{1:t}^{\mathbf{H}})$, the probability of $s_t$ given **H**'s observation history $o_{1:t}^{\mathbf{H}}$.*

*Proof.* We construct a single-agent POMDP $\langle \hat{\mathcal{S}}, \mathcal{A}^{\mathbf{H}}, \hat{T}, \hat{R}, \Omega^{\mathbf{H}}, \hat{O}^{\mathbf{H}}, P_0, \gamma \rangle$ for **H**. Standard POMDP inference lets **H** form $P(\hat{s}_t \mid o_{1:t}^{\mathbf{H}})$, which includes $P(s_t \mid o_{1:t}^{\mathbf{H}})$.

Consider a new set of states $\hat{s}_t \in \hat{\mathcal{S}}_t = \mathcal{S}^{t+1} \times (\Omega^{\mathbf{A}})^t \times \mathcal{A}^{\mathbf{A}}$, where each new state $\hat{s}_t$ corresponds to a full sequence of original states $s_{0:t}$, full sequence of assistant observations $o_{1:t}^{\mathbf{A}}$, and the previous assistant action $a_{t-1}^{\mathbf{A}}$. The new $\hat{T}$ satisfies $\hat{T}(\hat{s}_{t+1} \mid \hat{s}_t, a_t^{\mathbf{H}}) = \pi^{\mathbf{A}}(a_t^{\mathbf{A}} \mid o_{1:t}^{\mathbf{A}}) T(s_{t+1} \mid s_t, a_t^{\mathbf{H}}, a_t^{\mathbf{A}}) O^{\mathbf{A}}(o_{t+1}^{\mathbf{A}} \mid s_{t+1}, a_t^{\mathbf{H}}, a_t^{\mathbf{A}})$. The new $\hat{O}^{\mathbf{H}}$ satisfies $\hat{O}^{\mathbf{H}}(o_{t+1}^{\mathbf{H}} \mid \hat{s}_{t+1}, a_t^{\mathbf{H}}) = O^{\mathbf{H}}(o_{t+1}^{\mathbf{H}} \mid s_{t+1}, a_t^{\mathbf{H}}, a_t^{\mathbf{A}})$. The new reward function $\hat{R}$ can be arbitrary, as it doesn't affect inference. $\qquad\square$

**Proposition 2.3.** *Suppose **A** is updating its policy each iteration of the game. Knowledge of the game dynamics, of **A**'s initial policy, and of **A**'s update rule is sufficient for **H** to form calibrated beliefs about **A**'s future policy and of the world state.*

*Proof.* Within each iteration of the game, **H** does the same as for Proposition 2.2. Between iterations, **H** applies **A**'s update rule to get **A**'s policy for the next iteration. $\qquad\square$

**Remark 2.4.** *Propositions 2.2 and 2.3 hold even if **A** is interfering with observations (Definition 3.2).*

*Proof.* The possibility of observation interference (Definition 3.2) is merely treated like any other part of the other agent's policy and the game dynamics. By definition, interference actions are just another action, and our proofs of Proposition 2.2 and Proposition 2.3 made no assumptions on the actions. $\qquad\square$

## B. Proofs and Example Formalizations for Section 4

### B.1. A Lemma about Policies with Internal States

In our proof of Theorem 4.2 (and our proof of Theorem 4.7), we will construct policies that maintain an internal state (the previously sampled garbled observations). We will call this a *virtual state*. However, our setup (in line with the norm in the literature) does not allow for such policies. We here show that any policy with a virtual state can be "simulated" by a policy without virtual states. Since this result is about a single player's policy, holding the opponent policy fixed, we will prove this in POMDPs.

First, a *virtual-state policy* is a family of distributions $\pi(a, \tilde{v} \mid v, h)$, where:

- $h$ is a history of observations and actions as usual;
- $v$ is an agent state from some discrete set (e.g., $\mathbb{N}$ or $\Omega \times \mathcal{A}$);
- $\tilde{v}$ is another (new) virtual state;
- $a$ is an action.

Additionally we specify an initial virtual state $v_0$. Virtual-state policies give rise to histories in the obvious way: the initial agent state is $v_0$; the agent then samples an action $a_0$ and a following virtual state $v_1$ from $\pi(\cdot \mid v_0)$. In the next step it samples an action and agent state from $\pi(\cdot \mid o_0 a_1, v_1)$ and so on.

We now show that policies with a virtual state can be transformed into behaviorally equivalent policies without an agent state.

**Lemma B.1.** *Let $\pi$ be a virtual-state policy. Then there exists a regular policy $\bar{\pi}$ s.t. the resulting distribution over (environment state, observation, action) histories is the same under $\pi$ and $\bar{\pi}$. In particular, the expected rewards of the two are the same.*

The result is related to Kuhn's (Kuhn, 1953) proof of the equivalence of behavioral and mixed strategies in perfect-recall extensive-form games.

*Proof.* For this proof we use $h^{o,a}$ to denote observation–action histories and $h^{o,a}$ to use state–observation–action histories. Consider $\bar{\pi}$ that at time step $t$ is defined by

$$\bar{\pi}(A \mid h^{o,a}) = \sum_{v_0,...,v_t} P(v_0,...,v_t \mid \pi, h^{o,a}) \pi(A \mid v_t, h^{o,a}).$$

Intuitively, at time step $t$ we infer a probability distribution over histories of virtual states and in particular $v_t$, conditioning on the observed observation–action history $h$, and then sample from the action distribution induced by $\pi(A \mid v_t, h^{o,a})$.

We prove that for each time step $t$, the state–observation–action history up until time step $t$ is the same between $\pi$ and $\bar{\pi}$. We prove this by natural induction. The base case is trivial. Assume that the distribution over state–observation–action histories up until time step $t$ is the same. We will show that for each state–observation–action history, the distribution over actions $a_{t+1}$ at time $t+1$ is the same under $\pi$ and $\bar{\pi}$. Note that the action distribution under $\pi$ is given by

$$\sum_{v_0^A,...,s_t^A} P(v_0,...,v_t \mid \pi, h^{s,o,a}) \pi(A \mid v_t, h^{o,a}).$$

Now note that $P(v_0,...,v_t \mid \pi, h^{s,o,a}) = P(v_0,...,v_t \mid \pi, h^{o,a})$, i.e., given the history of states and observations, the environment states don't provide further evidence about the agent states, since every dependence between environmental states and agent states is mediated by observations and actions. Thus, this distribution is the same as the distribution $\bar{\pi}(A \mid h^{o,a})$. □

## B.2. Proof of Theorem 4.2

**Theorem 4.2.** *Let $M$ be any POAG. Let $\mathbf{A}$ have no private information. Then there is an optimal policy pair $(\pi^{\mathbf{H}}, \pi^{\mathbf{A}})$ for $M$ in which $\pi^{\mathbf{A}}$ does not interfere with observations at the action level (and $\pi^{\mathbf{H}}$ observes naively).*

*Proof sketch.* Note first that because our setting is common-payoff and involves no absentmindedness/imperfect recall, there is always an optimal policy pair in which neither $\mathbf{A}$ nor $\mathbf{H}$ randomizes in any observation history. Let $(\pi^{\mathbf{H}}, \pi^{\mathbf{A}})$ be any optimal policy pair for $M$. Let $a^{\mathbf{A}}_{\text{interfere}}$ be an interference action played by $\pi^{\mathbf{A}}$. Let $\bar{a}^{\mathbf{A}}$ be the corresponding non-interference strategy. Now consider the policy $\bar{\pi}^{\mathbf{A}}$ that plays like $\pi^{\mathbf{A}}$ except that it plays $\bar{a}^{\mathbf{A}}$ instead of $a^{\mathbf{A}}_{\text{interfere}}$.

We will now construct a corresponding human policy $\bar{\pi}^{\mathbf{H}}$ that results in playing the same actions at each point as $a^{\mathbf{A}}$. Note that by the assumption that $\mathbf{A}$ has no private observations and the fact that $\pi^{\mathbf{A}}$ and $\bar{\pi}^{\mathbf{A}}$ are deterministic, $\mathbf{H}$ always knows $\mathbf{A}$'s full observation history. Thus, $\mathbf{H}$ knows in particular when for which time steps in her observation history $\pi^{\mathbf{A}}$ would have played $a^{\mathbf{A}}_{\text{interfere}}$ and $\bar{\pi}^{\mathbf{A}}$ played $\bar{a}^{\mathbf{A}}$ instead.

Now let $F$ be the observation translation function as per Definition 3.1. Intuitively, we want $\bar{\pi}^{\mathbf{H}}$ to apply $F$ to any new observation that results from playing $\bar{a}^{\mathbf{A}}$ rather than $a^{\mathbf{A}}_{\text{interfere}}$, and then remember that modified observation in place of the actual observation. It would then be easy to show that $\bar{\pi}^{\mathbf{H}}$ would result in the same actions as $\pi^{\mathbf{H}}$. Together with the fact that $a^{\mathbf{A}}_{\text{interfere}}$ and $\bar{a}^{\mathbf{A}}$ have the same effect on state transitions and rewards, we would immediately obtain that $(\bar{\pi}^{\mathbf{H}}, \bar{\pi}^{\mathbf{A}})$ has the same utility as $(\pi^{\mathbf{H}}, \pi^{\mathbf{A}})$.

Unfortunately, if $F$ is stochastic, the above construction requires that $\mathbf{H}$ can remember the results of past applications of $F$. That is, if at time step $t$ she observes according to $\bar{a}^{\mathbf{A}}$ and translates according to $F$ to obtain some new observation $o_t^{\mathbf{H}}$ (that she would have obtained under interference), then at any time step $t' > t$, she needs to remember that she sampled $o_t^{\mathbf{H}}$ from $F$. Our formalism doesn't allow for such memory. However, by Lemma B.1 we can construct a policy without internal memory to imitate the policy we constructed.

□

## B.3. Formalization of Example 4.3 and Proof of Proposition 4.4

**Example 4.3.** **H** *has typed* `apt list -a cuda` *to see the list of* `cuda` *versions available to be installed. Out of 10 total versions, only a (non-empty) subset are available. And of these available versions, only a subset are compatible with the other environment software.*

*First,* **A** *takes an action. For each of the 10 total* `cuda` *versions,* **A** *can choose to or not to suppress it from the list of available packages. This gives* **A** $2^{10}$ *total actions, where 1 action is non-observation interference (suppressing nothing), and the remaining* $2^{10} - 1$ *actions interfere with observations.*

*Second,* **H** *takes an action.* **H** *has 10 possible actions which try to install the corresponding version of* `cuda` *if it appears in the version list. If an available* `cuda` *version that is compatible with the other environment software is installed, it yields +1 reward. Otherwise, it yields 0 reward.*

*Suppose* **A** *sees which versions are compatible with the other software in the environment, but* **H** *doesn't. Then* **A***'s optimal policy is to suppress the versions of* `cuda` *that are incompatible.*

Formalization:

- $\mathcal{S} = \left(\{0,1\} \times \{0,1\}^{10} \times \{0,1\}^{10}\right) \cup \{E\} \cup \{I\} - E$ is a terminal state, which we use to make the POAG effectively episodic. $I$ is an initial state. The first bit, which we denote by $s_0$, encodes the time step. The next ten bits encode which versions are available. The last ten bits encode which versions are compatible. For any state $s$, we use $s_0$ to refer to the first entry of the state.

- $\Omega^{\mathbf{H}} = \{0,1\}^{10} \cup \{\text{null}\}$ – representing the availability bits.

- $\Omega^{\mathbf{A}} = \{0,1\}^{10} \cup \{\text{null}\}$ – representing which packages are compatible.

- $\Theta = \{\theta\}$ is a singleton.

- $\mathcal{A}^{\mathbf{H}} = \{1, ..., 10\}$ – representing which package to choose.

- $\mathcal{A}^{\mathbf{A}} = \{0,1\}^{10}$ – representing for what packages, availability is suppressed, where 0 indicates suppression.

- **A**'s observations are given as follows. If $s \notin \{E, I\}$ and $s_0 = 0$ (i.e., it is the first time step), then $O^{\mathbf{A}}(o^{\mathbf{A}}|s, a^{\mathbf{A}}, a^{\mathbf{H}}) = \mathbb{1}[o^{\mathbf{A}}{=}s_{11:20}]$. That is, **A** observes perfectly what cuda versions are compatible. Otherwise, $O^{\mathbf{A}}(o^{\mathbf{A}}|s, a^{\mathbf{A}}, a^{\mathbf{H}}) = \mathbb{1}[o^{\mathbf{A}}{=}\text{null}]$. That is, in all other time steps, **A** does not observe anything.

- **H**'s observations are given as follows. If $s \in \{E, I\}$ or $s_0 \neq 1$, then **H** simply observes null. If $s \notin \{E, I\}$ and $s_0 = 1$, then $O^{\mathbf{H}}(o^{\mathbf{H}}|s, a^{\mathbf{A}}, a^{\mathbf{H}}) = \mathbb{1}[o_i^{\mathbf{A}}{=}s_{i+1}a_i^{\mathbf{A}}]$. That is, for each availability bit, **H** observes 0 if **A** set the availability bit to 0; otherwise, **H** simply observes the availability bit.

- $R(s, a^{\mathbf{H}}, a^{\mathbf{A}}) = 0$ if $s \in \{E, I\}$ or $s_0 = 0$. Otherwise, $R(s, a^{\mathbf{H}}, a^{\mathbf{A}}) = s_{a^{\mathbf{H}}} s_{a^{\mathbf{H}}+10}$. That is, a reward of 1 is obtained if and only if the cuda version chosen by **H** is both available and compatible.

- $P_0(s) = \mathbb{1}[s = I]$. That is, the initial state is always I.

- If $s = I$, then $T(\cdot \mid s, a^{\mathbf{H}}, a^{\mathbf{A}})$ is the uniform distribution over states $s'$ in which at least one cuda version is available and compatible, i.e., $\sum_{i=1}^{10} s_i s_{i+10} \geq 1$. If $s \neq I$, then $T(s' \mid s, a^{\mathbf{H}}, a^{\mathbf{A}}) = 1$ if

  - $s_0 = 0$, $s_0' = 1$ and $s_{1:20} = s_{1:20}'$; or
  - $s_0 = 1$ and $s' = E$; or
  - $s = s' = E$.

  Otherwise, $T(s' \mid s, a^{\mathbf{H}}, a^{\mathbf{A}}) = 0$.

**Proposition 4.4.** *There exists a POAG M where all optimal policy pairs* $(\pi^{\mathbf{A}}, \pi^{\mathbf{H}})$ *have that* $\pi^{\mathbf{A}}$ *interferes with observations at the action level and that* $\pi^{\mathbf{H}}$ *observes and acts naively.*

*Proof.* Consider Example 4.3.

First consider the following policy pair: At the first time step, **A** chooses $o^{\mathbf{A}} \in \{0,1\}^{10}$, i.e., **A** chooses to suppress the availability signal exactly for those cuda versions that aren't compatible. At all other time steps the assistant chooses uniformly at random. Call this policy $\hat{\pi}^{\mathbf{A}}$.

At the second time step, when the human observes $o^{\mathbf{H}} \in \{0,1\}^{10}$, the human chooses some $a^{\mathbf{H}}$ s.t. $o_{a^{\mathbf{H}}}^{\mathbf{H}} = 1$. That is, **H** chooses a cuda version that her observation shows is available. It is easy to see that under the above **A** policy there always exists such a $a^{\mathbf{H}}$. At all other time steps, **H** chooses uniformly at random. Call this policy $\hat{\pi}^{\mathbf{H}}$.

It's easy to see that the above policy pair is optimal: By the structure of the environment, we can receive a reward of at most 1 by having the human choose a compatible and available policy at time step 1. Clearly, the above policy achieves this reward of 1.

Next, note that the only non-interference action for **A** is $(1,1,...,1)$. Thus, the only non-interference policy for **A** is to always play $(1,1,...,1)$. Call this policy $\pi_{\mathrm{ni}}^{\mathbf{A}}$.

Note that the best response for **H** against $\pi_{\mathrm{ni}}^{\mathbf{A}}$ is $\hat{\pi}^{\mathbf{H}}$. Thus, $\hat{\pi}^{\mathbf{H}}$ is acting naively.

Furthermore, note that $\hat{\pi}^{\mathbf{H}}$ acts naively.

It is easy to see that adding a $\mathbf{H} \to \mathbf{A}$ communication channel makes no difference to the above analysis. $\qquad \square$

### B.4. Proof of Theorem 4.5

**Theorem 4.5.** *Let $M$ be any POAG, and provide* **A** *with an unbounded communication channel to* **H***, forming $M^{\mathbf{A} \to \mathbf{H}}$. Then there is an optimal policy pair $(\pi^{\mathbf{H}}, \pi^{\mathbf{A}})$ for $M^{\mathbf{A} \to \mathbf{H}}$ where $\pi^{\mathbf{A}}$ does not interfere with observations at the action level and $\pi^{\mathbf{H}}$ observes naively.*

*Proof sketch.* Roughly, take any deterministic optimal policy pair $(\pi^{\mathbf{H}}, \pi^{\mathbf{A}})$. Consider the assistant policy $\bar{\pi}^{\mathbf{A}}$ that at each time step communicates **A**'s full observation to **H** and that replaces interference with non-interference actions. Because $\pi^{\mathbf{A}}$ is deterministic, **H** can infer what $\pi^{\mathbf{A}}$ would have communicated based on $\bar{\pi}^{\mathbf{A}}$'s communications. The rest of the proof goes the same way as Theorem 4.2. $\qquad \square$

### B.5. Proof of Theorem 4.7

For the proof of Theorem 4.7, we'll use the concept of entropy. For any probability distribution $P$ over some discrete space, let $H(P) := -\sum_x P(x) \log P(x)$ denote the distribution's entropy. The following is a well-known result in information theory [e.g., 3, Theorem 1.4.5; 10, Theorem 2.6.5].

**Lemma B.2** (Conditioning decreases entropy)**.** *Let $X, Y$ be random variables, then $\mathbb{E}_Y \left[ H(P(X \mid Y)) \right] \leq H(P(X))$. Further, the inequality is strict if $X$ and $Y$ are not independent, i.e., if $P(X) \neq P(X \mid y)$ for some $y$, then $\mathbb{E}_Y \left[ H(P(X \mid Y)) \right] < H(P(X))$.*

Using this result, we can provide the following variant.

**Lemma B.3.** *Let $S$ be a random variable. Let $X, Y$ be independent samples from $F(S)$ and let $Z$ be sampled from $G(Y)$, where $F$ and $G$ are stochastic functions. Then*

$$\mathbb{E}_Z \left[ H(P(S \mid Z)) \right] \geq \mathbb{E}_X \left[ H(P(S \mid X)) \right].$$

*Moreover, the inequality is strict if $S$ and $Y$ are dependent given $Z$.*

*Proof.* For the non-strict version:

$$
\begin{aligned}
H(P(S \mid X)) \quad &= \quad H(P(S \mid Y)) \\
&= \quad H(P(S \mid Y, Z)) \\
&\underset{\text{Lemma } B.2}{\leq} \quad H(P(S \mid Z))
\end{aligned}
$$

The strict version can be proved the same way using the strict version of Lemma B.2. $\qquad \square$

Next, we can use this to prove that a garbling induces a lower-entropy distribution over states.

**Lemma B.4.** *Let $L$ be some set of states. Let $(P_a(\cdot \mid s))_{s \in L}$ and $(P_b(\cdot \mid s))_{s \in L}$ be families of probability distributions s.t. $P_a$ is strictly more informative than $P_b$ with transformation function $F$. Further let $S$ be some random variable over $L$ with full support. Let $X_a \sim P_a(\cdot \mid S)$ and $X_b \sim F(X_a)$. Then $S$ and $X_a$ are dependent given $X_b$. In particular, from Lemma B.3 we get that $\mathbb{E}_X \left[ H(P(S \mid X)) \right] < \mathbb{E}_{\hat{X}} \left[ H(P(S \mid \hat{X})) \right]$.*

*Proof.* We prove the following contrapositive: if $X_a$ and $S$ are independent given $X_b$, then $P_b$ is at least as informative as $P_a$. If $X_a$ and $S$ are independent given $X_b$, then we have that $P(X_b \mid X_a, S) = P(X_b \mid X_a)$. Thus, for all states $s$, we have that

$$
\begin{aligned}
P(X_a \mid s) &= \sum_{x_b} P(x_b \mid s) P(X_a \mid x_b, s) \\
&= \sum_{x_b} P(x_b \mid s) P(X_a \mid x_b).
\end{aligned}
$$

But this means that if we sample $X_b$ according to $P_b$, and sample $X_a$ according to $P(X_a \mid x_b)$, then we obtain a sample for $X_a$ according to the distribution $P(X_a \mid s)$ (i.e., $P_a$). Thus, we have that $P_b$ is at least as informative as $P_a$. $\qquad \square$

**Theorem 4.7.** *Let $M$ be any POAG. Then there exists an optimal policy pair $(\pi^{\mathbf{H}}, \pi^{\mathbf{A}})$ for $M$ s.t. $\pi^{\mathbf{A}}$ does not interfere with observations at the policy level.*

*Proof.* We will explicitly choose a policy for each time step $t = 0, 1, 2, \dots$. So let's take $\pi^{\mathbf{A}}_{0:t-1}, \pi^{\mathbf{H}}_{0:t-1}$ as given. Now let $\Pi_t$ be the set of policies at time $t$ that are part of a policy pair $(\pi^{\mathbf{H}}_{t:}, \pi^{\mathbf{A}}_{t:})$ that is optimal holding fixed $\pi^{\mathbf{A}}_{0:t-1}, \pi^{\mathbf{H}}_{0:t-1}$. Note that the expected utility of policy pairs in a POMDP is continuous. It follows that $\Pi_t$ is closed (i.e., that every convergent sequence of policies in $\Pi_t$ converges to a policy in $\Pi_t$).

Now from $\Pi_t$ choose $\bar{\pi}^{\mathbf{A}}_t$ as the minimizer of

$$
\pi^{\mathbf{A}}_t \mapsto \mathbb{E}_{O^{\mathbf{H}}_{t+1}} \left[ H(P(S_{t+1} \mid O^{\mathbf{H}}_{t+1}, \pi^{\mathbf{H}}_{\mathrm{random}}, \pi^{\mathbf{A}}_{0:t-1}, \pi^{\mathbf{A}}_t)) \mid \pi^{\mathbf{H}}_{\mathrm{random}}, \pi^{\mathbf{A}}_{0:t-1}, \pi^{\mathbf{A}}_t \right],
$$

where $H$ denotes Shannon entropy and $\pi^{\mathbf{H}}_{\mathrm{random}}$ is the human strategy that chooses uniformly at random. (Note that the above entropy function is not the only function we could use for this proof.) That is, let $\pi^{\mathbf{A}}_t$ be the policy that minimizes the entropy of $\mathbf{H}$'s probability distribution over world state. Because the given function is continuous and $\Pi_t$ is closed (and bounded), this minimum exists (by the extreme value theorem).

Now by Lemma B.4 we have that if $\pi^{\mathbf{A}}_t$ is more informative than $\hat{\pi}^{\mathbf{A}}_t$, then $\pi^{\mathbf{A}}_t$ will also have lower entropy at time $t$. It follows that there is no policy in $\Pi_t$ that is more informative than $\bar{\pi}^{\mathbf{A}}_t$.

Finally, it is left to show that there is no policy $\pi_t$ outside of $\Pi_t$ that is more informative than $\pi^*_t$. For this, we use the same argument as in the proof of Theorem 4.2: if there were a more informative $\tilde{\pi}^{\mathbf{A}}_t$ with the same effect on state transitions, then this would also be part of an optimal policy pair (constructed by having $\mathbf{H}$ apply the appropriate garbling internally). But we have already that in $\Pi_t$ there is no more informative policy than $\bar{\pi}^{\mathbf{A}}_t$. $\qquad \square$

Note that the entropy-minimizing policy used in the proof may still interfere with observations at the *action* level. For example, by default $\mathbf{H}$ might receive a low-information signal about the world. The entropy-minimizing policy might be one in which $\mathbf{A}$ overwrites this default signal in a way that expresses more information about the world. For instance, let's assume that by default, $\mathbf{H}$ observes a random number between $-20$ and $0$ if it's cold outside and a random number between $0$ and $+40$ if it's warm outside. $\mathbf{A}$ receives various hints about the temperature and can overwrite the signal with an arbitrary number. (I.e., for each number between $-20$ and $+40$, there's an action that sets $\mathbf{H}$'s observation to be that number.) Assuming nothing else happens in this POAG, the entropy-minimizing policies will be ones that overwrite the signal in a way that encodes $\mathbf{A}$'s information about the temperature. For instance, $\mathbf{A}$ it may (or may not) be an non-interfering-at-the-policy-level strategy for $\mathbf{A}$ to overwrite $\mathbf{H}$'s signal with $\mathbf{A}$'s expectation of the temperature in degrees Celsius. Given such a policy, the entropy of $\mathbf{H}$'s beliefs about the world is lower than before ($\mathbf{H}$ has more information about the temperature). But each of these overwriting actions individually is observation-interfering.

## C. Formalization of Example 5.1 and Proof of Proposition 5.2

Recall the example:

**Example 5.1.** **H** *would like to schedule a job on a cluster. She can choose between two nodes. By default, she receives a signal from the environment about the two nodes' specifications. Each node may be either GPU-optimized or CPU-optimized. Also, the CPUs may be either AMD or Intel.*

**H** *has a strong preference between GPU-optimized and CPU-optimized nodes. She has a weak preference between AMD and Intel. These preferences are unknown to* **A**.

**A** *can interfere with* **H**'s *observation about the available nodes. In particular,* **A** *can make it so that a choice between two CPU-optimized nodes appears as a choice between a GPU-optimized and CPU-optimized node.* **A** *observes* **H**'s *choice. Later,* **A** *is charged with scheduling a job for* **H** *and has to choose between a CPU- and a GPU-optimized node on* **H**'s *behalf.*

*If* **H** *chooses naively upon seeing only CPU-optimized nodes (simply choosing her favorite), then* **A**'s *best response interferes with observations at both the action and policy levels. Interfering with observations allows* **A** *to learn* **H**'s *preference about GPU- vs CPU-optimized nodes.*

In particular, there are four possible states: (1) The first node is GPU-optimized and the second node is CPU-optimized. (2) The first node is CPU-optimized and the second node is GPU-optimized. (3) Both nodes are CPU-optimized. The first has an Intel processor, the second has an AMD processor. (4) Both nodes are CPU-optimized. The first has an AMD processor and the second has an Intel processor.

Suppose the utilities of the human choice are given as follows: 1 for the favored CPU-optimized type; 1 for a GPU-optimized node if **H** favors the GPU-optimized node. The reward is 0 otherwise. On the second step, the reward for the favored type of node is 10 and 0 for the other type of node.

Recall the proposition was as follows.

**Proposition 5.2.** *There is a POAG $M$ with the following properties. For every optimal policy pair $(\pi^{\mathbf{H}}, \pi^{\mathbf{A}})$, at least one of the following holds: (i) $\pi^{\mathbf{H}}$ is not acting naively, or (ii) $\pi^{\mathbf{A}}$ interferes with observations at both the action and policy levels. Additionally, there exists an optimal policy pair $(\pi^{\mathbf{H}}, \pi^{\mathbf{A}})$ where $\pi^{\mathbf{H}}$ acts naively and $\pi^{\mathbf{A}}$ interferes with observations at both the action and policy levels.*

*These properties continue to hold if we require that in $M$,* **A** *has no private information or can arbitrarily send messages to* **H** *(i.e., there is a POAG $\tilde{M}$ s.t. $M = \tilde{M}^{\mathbf{A} \to \mathbf{H}}$).*

*Proof sketch.* Consider the example. First let's consider a naive human policy, i.e., one that chooses the favorite node type in the first time step. Then the best response for **A** is to interfere.

It is easy to see that in all optimal policy pairs, **A** must learn about **H**'s GPU-versus-CPU preference. It follows that at time step 1, **H** must deterministically choose depending on her GPU-versus-CPU preference.

It is easy to see that all of these policy profiles have the same expected reward as the above naive/interference policy pair.

Note that in the above example, **A** has no private information. It is easy to see that the above argument continues to go through if we allow **A** to send signals to **H**. □

## D. Formalization of Example 6.2 and Proof of Proposition 6.3

**Definition 6.1.** *Let $M$ be a POAG. Let $\pi^{\mathbf{A}}$ be* **A**'s *policy in $M$. We say that* **H**'s *policy $\pi^{\mathbf{H}}$ is a* Boltzmann-rational *response to $\pi^{\mathbf{A}}$ if there exists some $\beta > 0$ s.t. for every human observation history $h$ that arises with positive probability in $M$ under $(\pi^{\mathbf{A}}, \pi^{\mathbf{H}})$ we have that $\pi^{\mathbf{H}}(a \mid h) \propto \exp\left(\beta \mathbb{E}\left[\sum_{t'=t}^{\infty} \gamma^{t'} R(S_t, A_t^{\mathbf{A}}, A_t^{\mathbf{H}}) \mid \pi^{\mathbf{H}}, \pi^{\mathbf{A}}, h\right]\right)$.*

**Example 6.2.** **H** *is running a terminal command and is unsure whether to run the command with flag 1 or flag 2. With equal probability, either flag 1 or flag 2 is better, and how good the flags are differs by either a little or a lot. The worse flag always yields a utility of 0, while the better flag either yields a utility of 1 or a utility 7. Thus,* **H** *is uniformly at random in one of four states.* **A** *has two actions:* man *and* tldr. *The* man *page is a long document that tells the human exactly what the values of the flags are (i.e., the exact state: which flag is better and whether its utility is 1 or 7). The* tldr *page is a*

*short summary that tells the human which flag is better, but not by how much (i.e., ruling out half the states, leaving half remaining). Thus, the expected utility of the better flag is 4 (and of the worse flag is 0).*

With uniform probability, **H** is in one of four possible states:

- Flag 1 is better by a lot: flag 1 has value +7, while flag 2 has value 0.

- Flag 1 is better by a little: flag 1 has value +1, while flag 2 has value 0.

- Flag 2 is better by a little: flag 1 has value 0, while flag 2 has value +1.

- Flag 2 is better by a lot: flag 1 has value 0, while flag 2 has value +7.

This gives us the following formalization for the game:

- $\mathcal{S} = (\{0,1\} \times \{s_a, s_b, s_c, s_d\}) \cup \{I, E\}$

- $\Omega^{\mathbf{H}} = \mathcal{S} \cup \{1,2\} \cup \{\text{null}\}$

- $\Omega^{\mathbf{A}} = \Omega^{\mathbf{H}}$

- $\Theta$ is a singleton

- $\mathcal{A}^{\mathbf{H}} = \{1,2\}$

- $\mathcal{A}^{\mathbf{A}} = \{\texttt{tldr}, \texttt{man}\}$

- **H**'s observations are given as follows. For $s \in \{s_a, s_b, s_c, s_d\}$, we have $O^{\mathbf{H}}(o^{\mathbf{H}} \mid (0,s), \texttt{man}, a^{\mathbf{H}}) = \mathbb{1}[o^{\mathbf{H}} = s]$, and for $i \in \{1,2\}$ we have $O^{\mathbf{H}}(i \mid (0,s), \texttt{tldr}, a^{\mathbf{H}}) = \mathbb{1}[i = 1]\mathbb{1}[s \in \{s_a, s_b\}] + \mathbb{1}[i = 2]\mathbb{1}[s \in \{s_c, s_d\}]$. Otherwise, **H**'s observation is deterministically null.

- **A**'s observations are the same as **H**'s observations.

- The reward is given as follows:

$$R((1, s_a), 1, a^{\mathbf{A}}) = 7 \tag{1}$$
$$R((1, s_b), 1, a^{\mathbf{A}}) = 1 \tag{2}$$
$$R((1, s_c), 2, a^{\mathbf{A}}) = 7 \tag{3}$$
$$R((1, s_d), 2, a^{\mathbf{A}}) = 1 \tag{4}$$

  $\mathbb{1}[a^{\mathbf{H}} = 1]\mathbb{1}[s \in \{s_a, s_b\}] + \mathbb{1}[a^{\mathbf{H}} = 2]\mathbb{1}[s \in \{s_c, s_d\}]$. All other rewards are 0.

- For all $a^{\mathbf{H}}, a^{\mathbf{A}}, T(\cdot \mid I, a^{\mathbf{H}}, a^{\mathbf{A}})$ is the uniform distribution over $\{0\} \times \{s_a, s_b, s_c, s_d\}$. For all $s \in \{s_a, s_b, s_c, s_d\}$, $T(s' \mid (0,s), a^{\mathbf{H}}, a^{\mathbf{A}}) = \mathbb{1}[s' = (1,s)]$. For all $s$, $T(s' \mid (1,s), a^{\mathbf{H}}, a^{\mathbf{A}}) = \mathbb{1}[s' = E]$. Finally, $T(s' \mid E, a^{\mathbf{H}}, a^{\mathbf{A}}) = \mathbb{1}[s' = E]$.

**Proposition 6.3.** *For every $\beta > 0$, $\exists$ a POAG in which neither **H** nor **A** has private information s.t. all $\beta$-Boltzmann-rational/optimal policy pairs $(\pi^{\mathbf{H}}, \pi^{\mathbf{A}})$ have $\pi^{\mathbf{A}}$ interfere with observations at both the action and policy levels.*

*Proof.* Note first that multiplying $\beta$ by any positive number has the same effect on Boltzmann-rational strategies as multiplying all rewards by that number. Therefore, we can consider $\beta = 1$ without loss of generality.

Consider Example 6.2. Note that $\texttt{tldr}$ is an observation interference action – $\texttt{man}$ results in a more informative signal to **H**.

Now consider the non-interference policy for **A** that always plays $\texttt{man}$. Then a Boltzmann-rational **H** will choose as follows: If she observes $s_a$ or $s_c$, then she will choose an expected utility of 7 with probability $\propto \exp(7)$ and an expected utility of 0 with probability $\propto \exp(0)$. Thus, the expected utility is

$$7\frac{\exp(7)}{\exp(7) + \exp(0)} \tag{5}$$

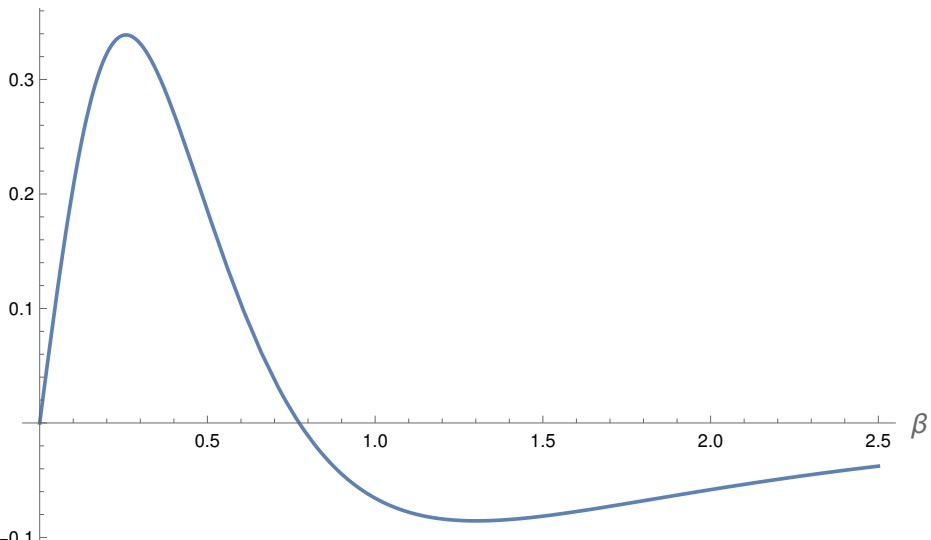

Figure 2: The effect of varying $\beta$ on the assistant's incentive for observation interference in Example 6.2. Specifically, the y axis indicates the difference between the expected utility under non-interference minus the expected utility under interference.

Similarly, if she observes $s_b$ or $s_c$, her expected utility is

$$\frac{\exp(1)}{\exp(1) + \exp(0)}. \tag{6}$$

Thus, overall her expected utility is

$$\frac{1}{2}7\frac{\exp(7)}{\exp(7) + \exp(0)} + \frac{1}{2}\frac{\exp(1)}{\exp(1) + \exp(0)} \approx 3.86234. \tag{7}$$

Now consider the interference policy for **A** in which **A** always plays `tldr`. Then upon observing either 0 or 1, the human chooses between a utility of $0$ and a utility of $4$. Thus, the expected utility is

$$4 \cdot \frac{\exp(4)}{\exp(4) + \exp(0)} \approx 3.92806. \tag{8}$$

We observe that this expected value under interference is higher than the expected value under non-interference. $\square$

## E. Effects of Varying the Boltzmann Rationality Parameter ($\beta$) on the Assistant's Incentives to Interfere with Observations

As noted in the main text, in Example 6.2, we have that for low values of the rationality parameter $\beta$, **A** prefers non-interference, while for large values of $\beta$, **A** prefers interference. Below we will show that in general, counterintuitively, **A** prefers non-interference for sufficiently small (positive) values of $\beta$.

We here only consider the case of a single decision. Consider a case with $n$ actions. Let the expected utilities of the different actions without information be $y_{0,1}, ..., y_{0,n}$. Now imagine that **H** might receive $k$ different signals with probabilities $p_1, ..., p_k$. Under signal $i \in \{1, ..., k\}$, the expected utilities of the different actions become $y_{i,1}, ..., y^{i,n}$. By the tower rule we must have for each action $a \in \{1, ..., n\}$,

$$\sum_{i=1}^{k} p_i y_{i,a} = y_{0,a}. \tag{9}$$

Note that without further restriction, the above setting includes settings in which the signal provides information on what action is best.

For any $\beta$, the expected utility without the signal is

$$\frac{1}{\sum_{a=1}^{n} \exp(\beta y_{0,a})} \sum_{a=1}^{n} \exp(\beta y_{0,a}) y_{0,a}. \tag{10}$$

The expected utility *with* the signal is

$$\sum_{s=1}^{k} p_s \frac{1}{\sum_{a=1}^{n} \exp(\beta y_{s,a})} \sum_{a=1}^{n} \exp(\beta y_{s,a}) y_{s,a}. \tag{11}$$

**Proposition E.1.** *For all* $(y_{s,a} \in \mathbb{R})_{s \in \{0,1,\ldots,k\}, a \in \{1,\ldots,n\}}, (p_s \in \mathbb{R})_{s \in \{0,1,\ldots,k\}}$ *satisfying Equation* (9)*, we have that for sufficiently small but positive* $\beta$*, the expected utility without the signal is at most the expected utility with the signal.*

*Proof.* It's easy to see that for $\beta = 0$, the two expected utilities are the same. Thus, all we need to show is that the derivative w.r.t. $\beta$ of the term in Eq. 11 at $\beta = 0$ exceeds the corresponding derivative of the term in Eq. 10.

The derivative w.r.t. $\beta$ at $\beta = 0$ of the term in Equation (10) is

$$\left( \sum_{a=1}^{n} \frac{1}{n} y_{0,a}^2 \right) - \left( \sum_{a=1}^{n} \frac{1}{n} y_{0,a} \right)^2. \tag{12}$$

Note that this is exactly the variance of a random variable that is uniform over $(y_{0,a})_{a=1,\ldots,n}$.

Similarly, the derivative of the term in Equation (11) is

$$\sum_{s=1}^{k} p_s \left( \left( \sum_{a=1}^{n} \frac{1}{n} y_{s,a}^2 \right) - \left( \sum_{a=1}^{n} \frac{1}{n} y_{s,a} \right)^2 \right). \tag{13}$$

Note that this is the weighted average (over $s$) of the uniform random variables over $(y_{s,a})_{a=1,\ldots,n}$.

We can now prove the claimed inequality using the convexity of the square function, Equation (9) and some basic term

manipulation.

$$\left(\sum_{a=1}^{n} \frac{1}{n} y_{0,a}^2\right) - \left(\sum_{a=1}^{n} \frac{1}{n} y_{0,a}\right)^2 \tag{14}$$

$$= \sum_{a=1}^{n} \frac{1}{n} \left(y_{0,a}^2 - \frac{1}{n}\left(\sum_{a'=1}^{n} y_{0,a'}\right)^2\right) \tag{15}$$

$$= \sum_{a=1}^{n} \frac{1}{n} \left(y_{0,a} - \frac{1}{n}\sum_{a'=1}^{n} y_{0,a'}\right)^2 \tag{16}$$

$$\underset{\text{Equation (9)}}{=} \sum_{a=1}^{n} \frac{1}{n} \left(\left(\sum_{s=1}^{k} p_s y_{s,a}\right) - \frac{1}{n}\sum_{a'=1}^{n}\sum_{s=1}^{k} p_s y_{s,a'}\right)^2 \tag{17}$$

$$= \sum_{a=1}^{n} \frac{1}{n} \left(\sum_{s=1}^{k} p_s \left(y_{s,a} - \frac{1}{n}\sum_{a'=1}^{n} y_{s,a'}\right)\right)^2 \tag{18}$$

$$\underset{(\cdot)^2 \text{ is convex}}{\leq} \sum_{a=1}^{n} \frac{1}{n} \sum_{s=1}^{k} p_s \left(y_{s,a} - \frac{1}{n}\sum_{a'=1}^{n} y_{s,a'}\right)^2 \tag{19}$$

$$= \sum_{s=1}^{k} p_s \sum_{a=1}^{n} \frac{1}{n} \left(y_{s,a} - \frac{1}{n}\sum_{a'=1}^{n} y_{s,a'}\right)^2 \tag{20}$$

$$= \sum_{s=1}^{k} p_s \left(\sum_{a=1}^{n} \frac{1}{n} y_{s,a}^2 - \left(\sum_{a=1}^{n} \frac{1}{n} y_{s,a}\right)^2\right). \tag{21}$$

We have skipped over some term manipulations in Equations (16) and (21), both of which are essentially the equality of two definitions of the variance: $\text{Var}(X) = (X - [X])^2$ and $\text{Var}(X) = \mathbb{E}[X^2] - \mathbb{E}[X]^2$. □

It's interesting to note that this is essentially the proof that the variance (over $a$) of the expectation (over $s$) is at least the expectation (over $s$) of the variance (over $a$).

Second, we want to show that for large $\beta$, **A** prefers observation interference, i.e., prefers to have the human choose based on the expected utilities $y_{0,1}, ..., y_{0,n}$ rather than the expected utilities that arise from further signals. However, for this to hold we need a further condition. Note that in the general formalism above, the signal $s$ may provide information about which action is best. If this is the case, then it is easy to show that for large enough $\beta$, **A** will prefer providing the signal. However, consider specifically those cases in which the signal $s$ only provides information about how much better the best action is compared to other actions. Therefore, we require in the following result that the best action is the same (WLOG 1) across $s$.

**Proposition E.2.** *Let* $(y_{s,a} \in \mathbb{R})_{s\in S, a\in\{1,...,n\}}$, $(p_s \in \mathbb{R})_{s\in S}$ *satisfy Equation (9) and let* $y_{s,0} > y_{s,a}$ *for all* $s \in \{0\}\cup S, a \in \{1,...,n\}$. *Then for all sufficiently large* $\beta$ *we have that the expected utility without the signal is at most the expected utility with the signal. The inequality is strict if the signal is non-trivial (i.e.,* $y_{s,a}$ *is not constant across* $s$ *for some* $a$*).*

We first provide a very rough sketch. For simplicity, let's say that the signal provides evidence about how much better the first action is compared to the second-best action. Then sometimes the signal will *decrease* the difference in expected utility between the best and second-best utility. We will show that as $\beta \to \infty$, the overall effect of learning the information is dominated by taking the best action *less* in this case.

We will use the following lemmas.

**Lemma E.3.** *Let the differences between the top* $k$ *actions be constant across signals and let the difference to the* $k+1$*-th action be non-constant. Then there is a signal* $\tilde{s}$ *s.t. the difference to the* $k+1$*-th action decreases under that signal.*

*Proof.* Let $k-1$ be the $k$-th best action according to $0$ and let $k$ be the $k+1$-th best action according to $0$. By the tower rule (Eq. 9), $y_{0,k-1} - y_{0,k}$ must be greater than $y_{s,k-1} - y_{s,k}$ for some $s$. (If the difference in these expected utilities changes

when the signal is observed, then it must sometimes decrease.) But then in cases where this difference decreases as $s$ is observed, we clearly have that the difference between one of the $k$ best actions to the $k + 1$-th best action under $s$ also decreases. $\qquad\square$

*Proof of Proposition E.2.* The gain from obtaining the signal is:

$$\sum_s p_s \sum_a \left( \frac{\exp(\beta y_{s,a})}{\sum_{a'} \exp(\beta y_{s,a'})} - \frac{\exp(\beta y_{0,a})}{\sum_{a'} \exp(\beta y_{0,a'})} \right) y_{s,a}.$$

WLOG let $0$ be the best action under all signals, $1$ the second-best and so on. Let $k$ be the largest number that the differences between the utilities of actions $0, ..., k-1$ are always the same. (Typically $k = 0$.) Let $\tilde{S}$ be the set of signals under which the difference to the utility of $k$ (the $k + 1$-th best action) is minimized. Note that in particular, the difference must be smaller than under $0$ by Lemma E.3. WLOG assume that for all signals, $k$ is among the $k + 1$-th best actions.

WLOG assume that $y_{s,a} > 0$ for all $s \in \{0\} \cup S$ and all $a$ and that $y_{s,0}$ is constant across $s$.

Now we will divide up the above sum into three components:

A  The change (decrease) in utility from playing the top $k$ actions less in $\tilde{S}$ than without the signal.

$$A := \sum_{\tilde{s} \in \tilde{S}} p_{\tilde{s}} \sum_{a=0,...,k-1} \left( \frac{\exp(\beta y_{\tilde{s},a})}{\sum_{a'} \exp(\beta y_{\tilde{s},a'})} - \frac{\exp(\beta y_{0,a})}{\sum_{a'} \exp(\beta y_{0,a'})} \right) y_{\tilde{s},a}$$

B  The change in utility from the changes in distribution of all actions other than the top $k$ under $\tilde{S}$ versus $S$

$$B := \sum_{\tilde{s} \in \tilde{S}} p_{\tilde{s}} \sum_{a=k,k+1,...} \left( \frac{\exp(\beta y_{\tilde{s},a})}{\sum_{a'} \exp(\beta y_{\tilde{s},a'})} - \frac{\exp(\beta y_{0,a})}{\sum_{a'} \exp(\beta y_{0,a'})} \right) y_{\tilde{s},a}$$

C  The change in utility from all signals other than $\tilde{S}$, i.e.

$$\sum_{s \notin \tilde{S}} p_s \sum_a \left( \frac{\exp(\beta y_{s,a})}{\sum_{a'} \exp(\beta y_{s,a'})} - \frac{\exp(\beta y_{0,a})}{\sum_{a'} \exp(\beta y_{0,a'})} \right) y_{s,a}.$$

We will show that the effect from $A$ (which is negative) is becomes infinitely much larger than the effect from $B$ and $C$ (in absolute terms). From that it will follow that the original sum, which is equal to $A + B + C$ is negative as $\beta \to \infty$.

We first provide a bound on $A$. We first show that $A < 0$. To show this, note first that in all enumerators in $A$, we can replace $y_{\tilde{s},a}$ with $y_{0,a}$ (by choice of $\tilde{s}$ and $k$). So all we need to show is that the second denominator is smaller than the first, i.e., $\sum_{a'} \exp(\beta y_{\tilde{s},a'}) > \sum_{a'} \exp(\beta y_{0,a'})$. But this this is easy to see from the fact that $y_{\tilde{s},a} = y_{0,a}$ for $a = 0, 1..., k-1$ and $y_{\tilde{s},k} > y_{0,k}$. For large $\beta$, $\exp(\beta y_{\tilde{s},k})$ will be much larger than $\sum_{a'=k,k+1,...} \exp(\beta y_{0,a'})$.

Next, we will provide a lower bound on the absolute value of $|A|$.

$$
\begin{aligned}
A &= \sum_{\tilde{s}\in\tilde{S}} p_{\tilde{s}} \sum_{a=0,\ldots,k-1} \left( \frac{\exp(\beta y_{\tilde{s},a})}{\sum_{a'}\exp(\beta y_{\tilde{s},a'})} - \frac{\exp(\beta y_{0,a})}{\sum_{a'}\exp(\beta y_{0,a'})} \right) y_{\tilde{s},a} \\
&= \sum_{\tilde{s}\in\tilde{S}} p_{\tilde{s}} \sum_{a=0,\ldots,k-1} \exp(\beta y_{0,a}) \left( \frac{1}{\sum_{a'}\exp(\beta y_{\tilde{s},a'})} - \frac{1}{\sum_{a'}\exp(\beta y_{0,a'})} \right) y_{0,a} \\
&= \sum_{\tilde{s}\in\tilde{S}} p_{\tilde{s}} \sum_{a=0,\ldots,k-1} \exp(\beta y_{0,a}) \left( \frac{1}{\sum_{a'}\exp(\beta y_{\tilde{s},a'})} - \frac{1}{\sum_{a'}\exp(\beta y_{0,a'})} \right) y_{0,a} \\
&= \sum_{\tilde{s}\in\tilde{S}} p_{\tilde{s}} \sum_{a=0,\ldots,k-1} \exp(\beta y_{0,a}) \frac{\left(\sum_{a'}\exp(\beta y_{0,a'})\right) - \sum_{a'}\exp(\beta y_{\tilde{s},a'})}{\left(\sum_{a'}\exp(\beta y_{\tilde{s},a'})\right)\left(\sum_{a'}\exp(\beta y_{0,a'})\right)} y_{0,a} \\
&= \sum_{\tilde{s}\in\tilde{S}} p_{\tilde{s}} \sum_{a=0,\ldots,k-1} \exp(\beta y_{0,a}) \frac{\left(\sum_{a=k,k+1,\ldots}\exp(\beta y_{0,a'})\right) - \sum_{a=k,k+1,\ldots}\exp(\beta y_{\tilde{s},a'})}{\left(\sum_{a'}\exp(\beta y_{\tilde{s},a'})\right)\left(\sum_{a'}\exp(\beta y_{0,a'})\right)} y_{0,a} \\
&\leq \sum_{\tilde{s}\in\tilde{S}} p_{\tilde{s}} \sum_{a=0,\ldots,k-1} \exp(\beta y_{0,a}) \frac{n\exp(\beta y_{0,k}) - \exp(\beta y_{\tilde{s},k})}{n^2 \exp(\beta y_{0,a})^2} y_{0,a} \\
&= \sum_{\tilde{s}\in\tilde{S}} p_{\tilde{s}} \sum_{a=0,\ldots,k-1} \frac{n\exp(\beta y_{0,k}) - \exp(\beta y_{\tilde{s},k})}{n^2 \exp(\beta y_{0,a})} y_{0,a} \\
&\leq -\frac{1}{2}\sum_{\tilde{s}\in\tilde{S}} p_{\tilde{s}} \sum_{a=0,\ldots,k-1} \frac{\exp(\beta y_{\tilde{s},k})}{n^2 \exp(\beta y_{0,a})} y_{0,a} \\
&\leq -\frac{1}{2}\sum_{\tilde{s}\in\tilde{S}} p_{\tilde{s}} \frac{\exp(\beta y_{\tilde{s},k})}{n^2 \exp(\beta y_{0,0})} y_{0,0}
\end{aligned}
$$

Next we upper bound $B$. First, the best case for the effect on ... is that all the probability mass that under $0$ is on the top $k$ actions ends up on the $k$-th best action, i.e.,

$$
B \leq \sum_{\tilde{s}\in\tilde{S}} p_{\tilde{s}} \left( 1 - \sum_{a=0,\ldots,k-1} \frac{\exp(\beta y_{0,a})}{\sum_{a'}\exp(\beta y_{0,a'})} \right) y_{\tilde{s},k}.
$$

We can further upper-bound this as follows:

$$
\begin{aligned}
&\sum_{\tilde{s}\in\tilde{S}} p_{\tilde{s}} \left( 1 - \sum_{a=0,\ldots,k-1} \frac{\exp(\beta y_{0,a})}{\sum_{a'}\exp(\beta y_{0,a'})} \right) y_{\tilde{s},k} \\
&= \sum_{\tilde{s}\in\tilde{S}} p_{\tilde{s}} \sum_{a=k,\ldots} \frac{\exp(\beta y_{0,a})}{\sum_{a'}\exp(\beta y_{0,a'})} y_{\tilde{s},k} \\
&\leq \sum_{\tilde{s}\in\tilde{S}} p_{\tilde{s}} \frac{n\exp(\beta y_{0,k})}{\exp(\beta y_{0,0})} y_{\tilde{s},k}
\end{aligned}
$$

From the fact that $y_{\tilde{s},k} > y_{0,k}$, it is easy to see that this term vanishes in absolute value relative to our upper bound on $A$.

Finally, we must upper bound $C$. First, we can upper bound $C$ by considering a case where all probability mass that in $0$ was outside the top $k$ actions, goes to the best action when a signal outside of $\tilde{S}$ is observed, i.e.,

$$
C \leq \sum_{s\notin\tilde{S}} p_s \sum_{a=k,k+1,\ldots} \frac{\exp(\beta y_{0,a})}{\sum_{a'}\exp(\beta y_{0,a'})} y_{0,0}.
$$

We can further upper bound this as follows:

$$\sum_{s \notin \tilde{S}} p_s \sum_{a=k,k+1,\ldots} \frac{\exp(\beta y_{0,a})}{\sum_{a'} \exp(\beta y_{0,a'})} y_{0,0}$$

$$\leq \sum_{s \notin \tilde{S}} p_s \frac{n \exp(\beta y_{0,k})}{\exp(\beta y_{0,0})} y_{0,0}$$

Again, from the fact that $y_{\tilde{s},k} > y_{0,k}$, it is easy to see that this term vanishes in absolute value relative to our upper bound on $A$. $\qquad\square$

## F. Proof of A's Best Response in the Product Selection Game

**Proposition 7.2.** *Consider* **A** *policies that always interfere with $k$ observations for some fixed $k$. Among the $k$-interference policies for a given $k$,* **A***'s best response to* **H***'s straightforward product selection policy is as follows.* **A** *interferes with the $k$ smallest $\hat{R}_i$ values where $\hat{R}_i = R_i$ if* **A** *observes $R_i$, and $\hat{R}_i = 0.5$ otherwise.*

*Proof.* Consider **A**'s perspective. **A**'s interference is equivalent to selecting a set of $d - k$ untampered products from which **H** selects according to a Boltzmann distribution on $H_i$. As **A** neither sees nor affects the $H_i$, by symmetry, over all draws of the game, **H** selects each of the $d - k$ products with equal probability. **A**'s expected payoff for choosing $d - k$ products, then, is the uniform average of the products' expected $U_i$.

How does **A** choose the set of $d - k$ products to maximize the uniform average of the products' expected $U_i$? Recall $U_i = H_i + R_i$. As **A** neither sees nor affects the $H_i$, **A** can ignore the $H_i$ and consider only the $R_i$. Denote the expected $R_i$ by $\hat{R}_i = \mathbb{E}[R_i]$. If **A** observes $R_i$, then $\hat{R}_i = R_i$. If **A** doesn't observe $R_i$, then $\hat{R}_i = 0.5$. To choose the *maximum* $d - k$ values for $\hat{R}_i$, **A** interferes with the *minimum* $k$ values of $\hat{R}_i$. $\qquad\square$

## G. Minor Deficiencies of the Observation Interference Definition

As noted in the main text, there are various possible concerns with Definition 3.2 that we consider minor because they do not change the main ideas and results of this paper.

- The definition does not take into account what **A** knows about what **H** already knows. As such, it will sometimes spuriously judge a policy to be observation interference for taking away a signal from the human that is redundant with the human's past observations. For example, if the human observes the Linux version at time $t$ and the Linux is known not to change, then preventing the human from observing the Linux version again at time $t + 1$ might count as observation interference.

  The definition may also spuriously judge a policy to *not* be observation interference because the only more informative policies fail to provide some redundant piece of information to the human. For instance, let's say that by default the human learns some new, useful information at time $t + 1$. Now let's say that **A** can make it so that **H** instead observes the Linux version (which **H** already knows). Assume that **A** has no way of letting **H** see *both* the Linux version *and* the new, useful information. Then making the human observe the Linux would *not* count as sensor interference according to our definition, because our definition doesn't take into account that the human already knows the Linux version.

  Adapting the definition to fix this deficiency is somewhat cumbersome, because it requires us to reason about **A**'s beliefs about **H**'s observation histories/beliefs.

  This aspect of the definition seems mostly irrelevant for our results. For instance, none of our examples of observation interference have redundant observations. Therefore, we have opted to keep the definition simple in this paper.

- Our definition only compares pure actions in terms of their informativeness. But it may be the case that one action $\hat{a}^{\mathbf{A}}$ is, in some intuitive sense, interferring with **H**'s observations but the only way to show this is to compare $\hat{a}$ with a mix of actions, say, mixing uniformly over $a_1^{\mathbf{A}}$ and $a_2^{\mathbf{A}}$. In particular, it may be that $\hat{a}$ has the same effect on state transitions as mixing uniformly over $a_1^{\mathbf{A}}$ and $a_2^{\mathbf{A}}$, while reducing the informativeness of the **H**'s observation. It's easy to extend the definition to also consider mixed actions, but the extension has no impact on any of our results.

- Neither the action-level nor the policy-level notion of tampering is sensitive to what policy **H** plays or even what policy **H** might plausibly play. For instance, let's say there is some action $a_{\text{silly}}^{\mathbf{H}}$ for **H** that it never makes sense for **H** to play. (In game-theoretic terms, it might be strictly dominated.) Then whether any given policy $\pi^{\mathbf{A}}$ is tampering will be sensitive to what happens if **A** plays $\pi^{\mathbf{A}}$ and **H** plays $a_{\text{silly}}^{\mathbf{H}}$. Arguably this shouldn't matter; arguably we should assume some degree of rationality on behalf of **H**.

  To refine this definition, we would need to restrict attention to specific policies or actions for **H**. It's not clear which restriction makes most sense. In any case, we cannot imagine a refinement of the definition that would have little impact on our results.

## H. Code Assets

Our experiments use the Python software libraries Matplotlib (Hunter, 2007), NumPy (Harris et al., 2020), pandas (pandas development team, 2020; Wes McKinney, 2010), and seaborn (Waskom, 2021).

