# OpenReview forum: "Observation Interference in Partially Observable Assistance Games"
_ICML.cc/2025/Conference — ICML 2025 poster_

### Official Review · Reviewer_dvjv · 2025-03-10

**Overall Recommendation:** 3

**Summary:**

The paper studies POAG where human and AI assistant only have partial observations. It shows that an optimal assistant who aims to maximize human reward needs to take observation-interfering actions, defined as an action showing a subset of information to human, for 3 purposes:
1.	Communicate AI’s private information
2.	Query human’s private information
3.	Help Boltzmann-rational humans

## update after rebuttal
My assessment has not changed after the rebuttal from the authors -- I believe that this paper makes a useful contribution to the field.

**Claims And Evidence:**

Section 3 sets up POAG, section 4 supports purpose 1, section 5 supports purpose 2, section 6 supports purpose 3, section 7 runs an experiment evaluating the pros and cons of interfering for purposes 1 and 3.

**Essential References Not Discussed:**

NA

**Experimental Designs Or Analyses:**

The experiments in Section 7 were intuitive and easy to follow.

**Methods And Evaluation Criteria:**

The authors use a very simple but effective hypothetical example to illustrate the framework. There is no real-world data to evaluate the ideas.

**Other Comments Or Suggestions:**

1) major typo: line 232 if A can communicate with A H
2) minor typo: line 104 Lehrer extend(s), line 380 utilities and 1 and 0
3) p. 6, the paragraph that starts with "At first sight" is a bit unclear. Why present this unlikely human strategy?
4) I did not follow the logic of the values of the utilities (e.g., 1,0,4,7) on p. 7, the paragraph that starts with "Intuitively, both the tldr and man pages allow the human to choose optimally?

**Other Strengths And Weaknesses:**

See comments above

**Questions For Authors:**

1) why didn’t the paper show an experiment for purpose 2?
2) For the experiment in section 7.1, there is no shared information correct? Would the presence of shared information change anything?

**Relation To Broader Scientific Literature:**

The paper provides a useful contribution to the literature by reviewing the conditions in which it is rational for AI to withhold information from a human user. This runs counter to the prevailing idea that the AI should reveal all available information. This paper might lead to interesting extensions where the AI might withhold not only information about the world state but also other information such as AI explanations (as this type of information often leads people to make inferior decisions).

There are some conceptual similarities of this work to off-switch games (this is not an essential reference though):
Hadfield-Menell, D., Dragan, A., Abbeel, P., & Russell, S. (2017). The off-switch game. In Workshops at the thirty-first aaai conference on artificial intelligence

**Theoretical Claims:**

The theoretical claims appear sound to me.

---

> ### Author Rebuttal · Authors · 2025-04-01
>
> # Thanks + response
>
> Thank you for your review. We are glad you think our work “provides a useful contribution to the literature” with technical details that are “sound,” “effective,” and “intuitive and easy to follow.” We appreciate the questions you ask about some of our examples and experiments.
>
> ## 1.
>
> > There are some conceptual similarities of this work to off-switch games (this is not an essential reference though): Hadfield-Menell, D., Dragan, A., Abbeel, P., & Russell, S. (2017). The off-switch game. In Workshops at the thirty-first aaai conference on artificial intelligence
>
> Thank you for sharing this reference. We agree it shares conceptual similarities, and we will cite it.
>
> **Action 1.1: We will add a citation to Hadfield-Menell’s et al.’s “The off-switch game.”**
>
> ## 2.
>
> > why didn’t the paper show an experiment for purpose 2?
>
> While we were limited by space constraints, we think this is an exciting direction for future work. Thus:
>
> **Action 2.1: We will discuss experiments for purpose 2 as a direction for future work in the “limitations and future work” section of our paper.**
>
> ## 3.
>
> > major typo: line 232 if A can communicate with A H
>
> We will fix this typo.
>
> > minor typo: line 104 Lehrer extend(s), line 380 utilities and 1 and 0
>
> We believe “Lehrer extend” is correct (see [rule 20 here](https://www.cs.columbia.edu/~hgs/etc/writing-bugs.html#:~:text=Note%20that%20%22et,%22shows%22.)) and will fix the line 380 typo.
>
> > p. 6, the paragraph that starts with "At first sight" is a bit unclear. Why present this unlikely human strategy?
>
> We were worried that Example 5.1 might be misconstrued as a counterexample to Theorem 4.2. We included this “At first sight” paragraph in an attempt to clarify why Example 5.1 is still consistent with Theorem 4.2.
>
> **Action 3.1: To clarify this, we will do the following:**
> * Remind the reader of the content of Theorem 4.2: “which states that non-interfering optimal policy pairs always exist”.
> * At the end of the “At first sight” paragraph, add: “The key point of Example 5.1 is therefore that---while there is _some_ optimal policy pair without observation interference---there is no plausible optimal policy pair that avoids observation interference. More specifically, we use the notion of acting naively (Definition 3.7) to express this notion of plausibility and rule out the above policy. We thus obtain the following proposition, which states that in some POAGs, if we want to play an optimal policy pair and we want H to be able to act naively, then A has to interfere with observations.”
>
> > I did not follow the logic of the values of the utilities (e.g., 1,0,4,7) on p. 7, the paragraph that starts with "Intuitively, both the tldr and man pages allow the human to choose optimally?
>
> We agree that the logic of the utilities for Example 6.2 should be clearer.
>
> **Action 3.2: To address this, we plan to discuss the utilities in Example 6.2 itself, rather than after it.**
> * In Example 6.2, we will spell it out in more detail as follows: “Specifically, the worse flag always yields a utility of 0. The better flag either yields a utility of 1 or a utility 7.”
> * We will also expand the parenthetical “i.e., the exact state: which flag is better and whether its utility is 1 or 7”.
> * And at the end of the example, write, “Therefore, the expected utility of the better flag is 4 (and the utility of the worse flag is 0).“
>
> > For the experiment in section 7.1, there is no shared information correct? Would the presence of shared information change anything?
>
> That is correct: there is no shared information. The presence of shared information would not necessarily change anything. As long as A has private information that is useful to the human, then it can still have an incentive to interfere with observations for purpose 1. And as long as A can make H’s decision easier by interfering, then A can still have an incentive to interfere with observations for purpose 3. This holds true even in the presence of shared information.
>
> ## What do you think?
>
> Thank you again for taking the time to review the paper and providing helpful feedback! **Do the above actions provide clarity to your questions?** If not, what further clarification or modifications could we make to improve your score?

---

### Official Review · Reviewer_SZMC · 2025-03-13

**Overall Recommendation:** 3

**Summary:**

This work studies a two player decentralised POMDP called Partially Observable Assistance Games. In this game, the authors study cases where it might be beneficial to one of the player (called assistant) to "interfere" with the observations of the other player (called human). They also identify situations, where such an undesirable interference should not happen (e.g. without any private information, or with a free communication channel).

For a different definition of observation interference though (at policy level vs action level), they provide a result guaranteeing the existence of interference free optimal policies.

Lastly, this work shows that observation interference might also be optimal in the case of irrational or 'naive' human player.

**Claims And Evidence:**

N/A

**Essential References Not Discussed:**

N/A

**Experimental Designs Or Analyses:**

N/A

**Methods And Evaluation Criteria:**

N/A

**Other Comments Or Suggestions:**

Studying situations where hiding some private information, even in collaborative games, is of strong interest in my opinion. I think this is even more general than this "assistance game" motivation.

My main concern is about the clarity of the provided mathematical framework. Notably, there are a lot of definitions, notations and objects at stake and it is hard to understand the real meaning of all of them.

As an example, the real difference between Definition 3.2 and Definition 4.6 is unclear to me (notably because it does not affect state transition), even after reading the illustrating example. In consequence, I don't understand why this shold lead to a significant difference in the optimal policy pairs, given by Theorem 4.7 vs Proposition 4.4.

After looking more carefully at the appendix, this seems more due to some subtleties in the definitions than a fundamental diferrence between these definitions.

Probably related is my impression that the POMDP framework might be overcomplicated with the messages that the authors want to give. Indeed, when looking at the mathematical definition of the example provided along the paper, this example corresponds to a degenerate POMDP (the stationary distribution cycles in a dummy terminal state). I thus feel that a much simpler problem formulation, e.g., in a normal form game, could have led to the same kind of conclusions. As a consequence, the messages of this paper are hard to grasp, due to an unnecessary (in my opinion) level of complexity in the setting, that requires the introduction of numerous definitions and notations.

Instead, I would have preferred a much simpler setting that draws similar conclusions.

Additionally, the example given along the paper indeed seems a necessary part to grasp all the presented concepts. However, I find this specific example unclear at some points, and it didn't really help me understand in the end (I particularly did not understand the example after Theorem 4.7).

Lastly, I am not sure of the relevance of this setting to RLHF, and I would need more explanation to really think this setting could apply to RLHF.

---------------------
# Minor comments

It seems in the paper that the assistance first takes his action, the human observes it and then takes action based on the assistance action. I did not see it clearly stated in the paper though.

Definition 2.1: please give the full name for DecPOMDP at least once

Line 132: for what utilities are they Nash Equilibria?

I don't get how Propositions 2.2 and 2.3 are actual propositions. They are just probabilistic statements (no need for a formal environment here in my opinion).

In definition 3.1, what is the definition for a stochastic function?

Line 231 (right): I think there is a typo, it should be "if A can communicate with H"

**Other Strengths And Weaknesses:**

N/A

**Questions For Authors:**

What is so specific to POMDP in the final conclusions drawn in this work? It seems to me that all the statements are somehow consequences of statements for stateless games (eg normal form game)

See other questions above

**Relation To Broader Scientific Literature:**

N/A

**Theoretical Claims:**

N/A

---

> ### Author Rebuttal · Authors · 2025-04-01
>
> # Thanks + response
>
> Thank you for your review. We are glad you find our setting “of strong interest,” and we appreciate your help with the clarity and simplicity of our paper.
>
> ## 1.
>
> > real difference between Definition 3.2 and Definition 4.6 is unclear… I did not understand the example after Theorem 4.7.
>
> Thank you - we want to be sure we clearly communicate these important points. We will:
>
> **Action 1.1: Extend our discussion on line 290 of the example after Theorem 4.7 to more thoroughly explain how Definition 4.6 differs from Definition 3.2.** The revised discussion will read:
>
> “We now revisit Example 4.3. For observation tampering under our earlier Definition 3.2, H simply knows that A has taken the *action* to suppress some versions of cuda. However, H does not know anything about A’s *policy*. For all H knows, A’s policy could be to randomly suppress cuda versions or to always suppress the same cuda version. Thus, suppressing any version is strictly less informative for H than the list of all available versions. This is why Definition 3.2 calls suppressing versions ‘tampering at the action level.’
>
> The key difference with Definition 4.6 is that H knows A’s policy. Suppose that A’s policy $\pi^\mathbf{A}$ is to suppress exactly the versions of cuda that are incompatible with the other software in the environment. Because H knows that A suppressed the incompatible cuda versions, seeing the filtered list tells H which versions of cuda are compatible! Although suppressing versions is strictly less informative under Definition 3.2 (when H doesn’t know A’s policy), suppressing versions provides H with new information under Definition 4.6 (when H knows A’s policy). Accordingly, $\pi^\mathbf{A}$ is interfering with observations at the action level *but not at the policy level*.”
>
> **Action 1.2: Move the discussion of the example after Theorem 4.7 (which is currently on Line 290) upward to line 280 so that it immediately follows Definition 4.6.** This way, the reader will immediately see an explanation of the intuition behind Definition 4.6 before Theorem 4.7 (which relies on Definition 4.6).
>
> **Question 1.3: Do these revisions provide clarity? If not, please let us know, and we will work to clarify further.**
>
> ## 2.
>
> > What is so specific to POMDP? … POMDP[s] might be overcomplicated…. A [much simpler] normal form game, could have led to the same conclusions.
>
> We require a framework with private information and sequential play, which normal-form games lack. Observation interference (both Definitions 3.2 and 4.6) needs A to choose first, then H to observe something dependent on A's choice.
>
> We study causes of observation interference which need:
> * For Communication (Section 4): A observes privately → A acts → H observes → H chooses (Example 4.1)
> * For Querying (Section 5): A acts → H observes → H acts → A observes → A acts (Example 5.1)
>
> While extensive-form games could use fewer "dummy" states, they're similarly complex as DecPOMDPs, which are standard in assistance games literature (Hadfield-Menell et al., 2016; Shah et al., 2020).
>
> **Question 2.1: Does this clarify why we can’t simplify our setup?**
>
> ## 3.
>
> > I would need more explanation to really think this setting could apply to RLHF.
>
> **Action 3.1: At the start of Section 4.1, we will add the following explanation:**
>
> “We can model RLHF within the POAG framework as follows:
> * The assistant’s goal in RLHF is to satisfy the human’s preferences. In a POAG, this corresponds to the shared reward function $R$ which has a parameterization $\theta$ that only H knows.
> * In RLHF, the assistant rolls out trajectories, and the human’s picks which trajectory is preferred. A POAG can model this by letting H observe pairs of trajectories explored by A but only giving H a binary action (to choose which trajectory H prefers).
> * A’s final RLHF policy maximizes an estimate of $R$ based on a dataset of H’s preference comparisons (Lang *et al.* (2024), Proposition 4.1). In the POAG framework, A can compute this policy based on A’s observations of H’s binary actions.”
>
> ## 4.
>
> For the minor comments, we will:
> * Add to line 115 “POAGs inherit the generality of DecPOMDPs: POAGs can model games where H acts first, where A acts first, or where H and A act simultaneously.”
> * Add the full name for DecPOMDP to Line 111.
> * Add a clarification so that line 132 reads: “... Nash equilibria for the shared reward function R.”
> * Add this definition of a stochastic function: “a function mapping observations to random variables over observations”.
> * Fix the line 231 typo.
>
> > how Propositions 2.2 and 2.3 are actual propositions
>
> Appendix A needs these propositions to have names in order to refer to them and prove them. To clarify, we will add “(See Appendix A for proofs.)” to line 148.
>
> ## What do you think?
>
> Thank you again for your review. **Do the above actions address your concerns?** If not, what further clarifications or modifications could we make to improve your score?

---

> > ### Comment · Reviewer_SZMC · 2025-04-03
> >
> > I thank the authors for their answer. The proposed modifications for the revised version will surely help in the clarity of the paper and especially in understanding its relation with RLHF.

---

### Official Review · Reviewer_fCAi · 2025-03-14

**Overall Recommendation:** 3

**Summary:**

Paper studies the conditions under which an agent has an incentive to perform observation interference (take an action which returns partial state observation to the human) even when the goals are aligned. The thought of taking such an action, at surface level seems adversarial and counter intuitive. However, the paper discusses conditions such as when the human is making decisions based on immediate reward, when agent wants to reveal some private information, or when the human is irrational and restricting observation is a way of forcing them to be rational. All of these comes at a cost of destroying state information which causes the trade off.

**Claims And Evidence:**

yes, the paper is great to read.

**Essential References Not Discussed:**

discussed.

**Experimental Designs Or Analyses:**

yes.

**Methods And Evaluation Criteria:**

yes.

**Other Comments Or Suggestions:**

see above comments.

**Other Strengths And Weaknesses:**

I enjoyed reading the paper.

I have a few concerns :

1. What is the practical applicability of the work? Are there domains beyond the curated examples where results of the paper are be discussed? What is the main impact of these results in such domains?

2. What happens if A has multiple private information?

3. Are actions like, attempt to open door, for an agent trying to expose that the door is locked, a valid example for agent exposing private information, (let's say human could never have been in the room). If so, this action doesn't alter the environment but exposes private information. Therefore, in the presence of both such actions - observation interfering and signaling which would be preferred?

**Questions For Authors:**

see above comments.

**Relation To Broader Scientific Literature:**

yes, this is very relevant to assistant games.

**Theoretical Claims:**

yes, and the general reasoning discussed in the paper is helpful to build intuition for the theory.

---

> ### Author Rebuttal · Authors · 2025-04-01
>
> # Thanks + response
>
> Thank you for your review. We are glad you consider our work “very relevant” to the broader scientific literature and that “the general reasoning discussed in the paper is helpful to build intuition for the theory.” We appreciate the questions you raise about the practical applicability of our work along with the example scenarios that you pose.
>
> ## 1.
>
> > What is the practical applicability of the work? Are there domains beyond the curated examples where results of the paper are be discussed? What is the main impact of these results in such domains?
>
> Thank you for raising these important questions about the practical applicability of our work. To address these questions:
>
> **Action 1.1: We will include a discussion of the practical applicability of our work in the “Impact Statement” section of our paper.** We propose the following text:
>
> “AI assistants are being developed and deployed in settings where humans can only partially observe what’s happening. For example, AI assistants including ChatGPT, Claude, and Gemini can search the web while only returning summaries to users ([OpenAI, 2024](https://openai.com/index/introducing-chatgpt-search/); [Anthropic, 2025](https://www.anthropic.com/news/web-search); [Google, 2024](https://blog.google/products/gemini/google-gemini-deep-research/)). Moreover, the sorts of AI models powering these assistants are processing increasingly long inputs. Whereas the original ChatGPT model could only process 4096 input tokens, today’s Gemini 1.5 Pro can process 2,000,000 input tokens—which is roughly 100,000 lines of code, or 16 novels of average length in English ([Google, 2025](https://ai.google.dev/gemini-api/docs/long-context)). In the future, we anticipate that AI assistants will be deployed at increasing scale, independently taking more actions on behalf of users and processing increasingly long context lengths. We thus expect that over time, humans will have less and less ability to directly observe everything that’s happening.
>
> The goal of our work is to lay a theoretical foundation for understanding when AI assistants have an incentive to interfere with human observations. Our results create a nuanced picture, suggesting that not all observation interference is inherently bad. In practice, we expect that AI assistants will exhibit observation interference for a mix of good and bad reasons. With this theory, our goal is to help practitioners disentangle these different incentives for observation interference when they emerge in practice.”
>
> ## 2.
>
> > What happens if A has multiple private information?
>
> While we prove all our “negative” results (results showing that observation tampering can happen) with simple examples in which both A and H only make a single observation once, all our positive results apply to any POAG, and thus allow settings in which both A and H observe multiple times!
>
> **Action 2.1: To clarify how general our setting is, we will add the following sentence to Section 2.1:**
>
> “While all the examples in this paper are quite simple, note that the POAG setup and thus all our positive results are very general, allowing both H and A to observe private information at multiple times, taking actions that influence both the state and each other’s observations, and so on.”
>
> ## 3.
>
> > Are actions like, attempt to open door, for an agent trying to expose that the door is locked, a valid example for agent exposing private information, (let's say human could never have been in the room). If so, this action doesn't alter the environment but exposes private information. Therefore, in the presence of both such actions - observation interfering and signaling which would be preferred?
>
> Thank you for proposing this great “locked door” example.
>
> **Action 3.1: We will add the locked door example to our paper after Definition 4.6, with this text:**
>
> “While in our examples, we will mostly consider actions that in some sense act directly on H’s observations, Definition 4.6 also considers the informational effects of physical actions. For example, if A (visibly) tries to open a door that A knows to be locked, then this reveals to H that the door is locked. Consequently, not trying to open the door (when A knows it to be locked) is an instance of observation interference in the sense of Definition 4.6. While having the same (null) effect on the state of the world, trying to open the door provides H with more information about the world.”
>
> ## What do you think?
>
> Thank you again for taking the time to review the paper and providing helpful feedback. **Do the above actions address your questions about the paper?** If not, what further clarification or modifications could we make to improve your score?

---

### Official Review · Reviewer_MYkQ · 2025-03-15

**Overall Recommendation:** 3

**Summary:**

This paper investigates observation interference by AI assistants in partially observable assistance games (POAGs), where both the AI and the human have limited information. The authors demonstrate that an optimal assistant may have incentives to interfere with observations to communicate private information, query human preferences when the human acts naively, and help the human make better decisions when the human is irrational. Defining observation interference as providing less informative signal, they show that while action-level interference may sometimes be necessary, policy-level interference is never required. This finding suggests that although observation interference involves sacrificing some information, it can benefit the human by facilitating the communication of more critical information. Experiments further explore the trade-offs influenced by the amount of the AI's private information and the degree of the human's rationality.

**Claims And Evidence:**

Please refer to Strengths And Weaknesses.

**Essential References Not Discussed:**

Please refer to Strengths And Weaknesses.

**Experimental Designs Or Analyses:**

Please refer to Strengths And Weaknesses.

**Methods And Evaluation Criteria:**

Please refer to Strengths And Weaknesses.

**Other Comments Or Suggestions:**

N/A

**Other Strengths And Weaknesses:**

**[Strengths]**
- This paper is well-organized and easy to follow, presenting formal definitions, solid proofs, examples, and experimental results.

- This paper addresses a crucial aspect of the human-AI value alignment problem by considering the more realistic scenario of partial observability. The theoretical analysis of observation interference and the three distinct incentives may have important implications for building trustworthy AI systems.

- Although the authors acknowledge some minor limitations in the definition of observation interference in Appendix G, the definitions, theoretical claims, and preliminary results appear mathematically sound.

**[Weaknesses]**
- The analysis of different incentives for observation interference often relies on specific assumptions about human behavior, such as naivety or adherence to the Boltzmann rationality model. The extent to which these assumptions accurately capture real human behavior across different contexts could affect the practical relevance of the findings. Additionally, the communication channels appear to assume that the human and the AI agent can perfectly understand each other, which may not hold in more complex scenarios.

- While the experimental model provides valuable insights, the paper would benefit from empirical validation with human subjects to test its theoretical predictions.

- The gap between theoretical optimality and practical implementation raises questions about the design of AI systems. A more in-depth discussion of how these insights could direct the development of AI assistants may enhance the paper.

**Questions For Authors:**

- Consider the computational complexity (NEXP-hard) of finding optimal policies in POAGs, what are your initial thoughts on how the theoretical insights from this paper could be practically applied to the design and implementation of real-world AI assistants, particularly those operating in complex and partially observable environments?

- Have you considered the ethical implications of designing AI systems that may optimally interfere with observations? What constraints might be appropriate to ensure that such interference does not lead to problematic patterns of interaction?

**Relation To Broader Scientific Literature:**

Please refer to Strengths And Weaknesses.

**Theoretical Claims:**

Please refer to Strengths And Weaknesses.

---

> ### Author Rebuttal · Authors · 2025-04-01
>
> # Thanks + response
>
> Thank you for your review. We are glad you consider our work to "address a crucial aspect of the human-AI value alignment problem" and "have important implications for building trustworthy AI systems." We appreciate the discussion points you raised.
>
> ## 1.
>
> > The analysis… relies on specific assumptions… [such as] the Boltzmann rational model.
>
> In this first paper on observation interference in human-AI interaction, we seek to lay a theoretical foundation using simple, well-known human models. Our results consider several models, including optimal play, decisions based on immediate outcomes, and Boltzmann rationality. Many published papers consider only the Boltzmann model, which is standard in the literature. For example, Hong Jun Jeon et al. (NeurIPS 2020) considers 11 feedback types that all assume Boltzmann rationality. Similarly, Ramachandran and Amir (IJCAI 2007) and Laidlaw and Dragan (ICLR 2022) exclusively study this model. We hope our paper will be judged by a consistent standard with other published work.
>
> **Action 1.1: In "Limitations and Future Work," we will add discussion of human modeling assumptions, suggesting additional models as a direction for future work.**
>
> ## 2.
>
> > the communication channels appear to assume that the human and the AI agent can perfectly understand each other
>
> We agree that communication is not a fully general solution to observation interference. But we don't view this as a weakness. If anything, our nuanced study would be less relevant if communication was a fully satisfying solution.
>
> **Action 2.1: We will add the following paragraph on practical limitations of unbounded communication channels after Theorem 4.5:**
>
> "One could argue that in practice, an unrestricted communication channel between A and H could usually be made available. However, Theorem 4.5 ignores various real-world obstacles. For one, it considers communication that incurs no cost, but realistically communication costs the human time and effort. Second, the optimal policy pair requires A to send information in a way that H can reliably understand and act on. We should expect that in practice, A and H sometimes cannot understand each other. Therefore, despite Theorem 4.5, we think observation interference is of broad practical relevance, even where A can, e.g., send text messages to H."
>
> ## 3.
>
> > the paper would benefit from empirical validation with human subjects
>
> We agree this would be valuable for future work, but we also believe that would be a different kind of paper. This first paper on the topic is about laying theoretical foundations.
>
> **Action 3.1: In "Limitations and Future Work," we will discuss empirical validation with human subjects as a valuable future direction.**
>
> ## 4.
>
> > Consider the computational complexity (NEXP-hard) of finding optimal policies in POAGs, how can the theoretical insights from this paper be practically applied to the real world?
>
> **Action 4.1: We will add the following discussion to our paper:**
>
> "How might our results apply given that finding optimal policies in POAGs is NEXP-hard?
>
> Most of our paper is descriptive, characterizing when observation tampering could happen. Complexity considerations could affect these results in either direction. It's easy to construct environments where finding good observation-interfering policies is computationally intractable but constructing good non-interfering policies is easy; and vice versa. In practice, complexities of the environment can be orthogonal to incentives to interfere. For instance, a real-world version of the CUDA example is complex (A assesses complicated software compatibility issues), but the decision whether to interfere with observations is easy. We believe our characterizations remain useful even in complex environments (where we can't expect optimal policies), although we can't make as definitive claims as we can about optimal policies.
>
> We have discussed allowing communication between H and A. A complexity-theoretic argument favors this solution: If H and A share all private information, the game effectively turns from a DecPOMDP into a POMDP. Solving POMDPs is PSPACE-complete and thus likely easier than solving DecPOMDPs."
>
> ## 5.
>
> > the ethical implications of designing AI systems that may optimally interfere… What constraints might be appropriate?
>
> Please see Action 1.1 in our response to Reviewer fCAi, where we discuss implications. Our theory reveals that optimal assistants must sometimes interfere with observations, so interference is a nuanced issue. Appropriate constraints might include transparency about when interference occurs, alignment with user goals, and user controls to override interference when desired. As you suggest, future work with human subjects is needed to refine these constraints in practice.
>
> ## What do you think?
>
> Thank you again. **Do the above actions address your concerns?** If not, what further changes could we make to improve your score?

---

### Decision · Program_Chairs · 2025-05-01

**Decision:**

Accept (poster)

**Comment:**

The reviewers agreed that the paper studies an interesting problem setting of partially observable assistance games, and that the findings could be potentially useful for human-AI alignment in real-world applications. However, the reviewers also raised several concerns and questions in their initial reviews. We want to thank the authors for their detailed responses, which helped in answering the reviewers’ key questions. The reviewers have an overall positive assessment of the paper, and there is a consensus for acceptance. The reviewers have provided detailed feedback, and we strongly encourage the authors to incorporate this feedback when preparing the final version of the paper.